# Synthetic biology-instructed transdermal microneedle patch for traceable photodynamic therapy

Gang He [1], Yashi Li[1], Muhammad Rizwan Younis[1], Lian-Hua Fu[1], Ting He[1], Shan Lei [1], Jing Lin[1] & Peng Huang [1] ✉

5-Aminolevulinic acid-based photodynamic therapy heavily depends on the biological transformation efficiency of 5-aminolevulinic acid to protoporphyrin IX, while the lack of an effective delivery system and imaging navigation are major hurdles in improving the accumulation of protoporphyrin IX and optimizing therapeutic parameters. Herein, we leverage a synthetic biology approach to construct a transdermal theranostic microneedle patch integrated with 5-aminolevulinic acid and catalase co-loaded tumor acidity-responsive copper-doped calcium phosphate nanoparticles for efficient 5-aminolevulinic acid-based photodynamic therapy by maximizing the enrichment of intratumoral protoporphyrin IX. We show that continuous oxygen generation by catalase in vivo reverses tumor hypoxia, enhances protoporphyrin IX accumulation by blocking protoporphyrin IX efflux (downregulating hypoxia-inducible factor-1α and ferrochelatase) and upregulates protoporphyrin IX biosynthesis (providing exogenous 5-aminolevulinic acid and upregulating ALA-synthetase). In vivo fluorescence/ photoacoustic duplex imaging can monitor intratumoral oxygen saturation and protoporphyrin IX metabolic kinetics simultaneously. This approach thus facilitates the optimization of therapeutic parameters for different cancers to realize $Ca^{2+}/Cu^{2+}$-interferences-enhanced repeatable photodynamic therapy, making this theranostic patch promising for clinical practice.

5-Aminolevulinic acid (5-ALA), the first heme precursor, can be selectively absorbed by cancer cells and rapidly transformed into its fluorescent form, protoporphyrin IX (PpIX), with unprecedented diagnostic accuracy[1–3]. Consequently, the photochemical reaction based on 5-ALA-photodynamic therapy (5-ALA-PDT) mainly occurs in cancer cells, avoiding the risk of off-target toxicity[4–6]. Although topical 5-ALA-PDT has become the mainstream treatment option for patients with actinic keratosis or basal cell carcinoma, the limited biosynthesis concentration of PpIX and poor tumor retention[3,7,8], tumor hypoxic microenvironment[9–11], and powerful antioxidant system[12–14] impede 5-ALA-PDT to treat solid malignant tumors. To address these problems,

it is critical to improve the biosynthetic kinetics of PpIX to make it beneficial for theranostics and reshape tumor microenvironment to attenuate the factors that are detrimental to 5-ALA-PDT[15–18].

Maximizing the enrichment of intratumoral PpIX is a prerequisite for 5-ALA-PDT[19–21]. Unfortunately, the high polarity and hydrophilicity of 5-ALA restrict its effective penetration in the stratum corneum and cell membrane through passive diffusion, resulting in relatively low bioavailability of 5-ALA via oral and transdermal administration[3,8,22,23]. The transformation efficiency of 5-ALA into PpIX is also crucial and determines the maximum enrichment time point (METP), tumor retention time, and physiological concentration of PpIX[7,22,24]. Based on

[1]Marshall Laboratory of Biomedical Engineering, International Cancer Center, Laboratory of Evolutionary Theranostics (LET), School of Biomedical Engineering, Shenzhen University Health Science Center, Shenzhen 518060, China. ✉e-mail: peng.huang@szu.edu.cn

the imbalance of PpIX and $Fe^{2+}$ supply caused by the Warburg effect of cancer cells[3,7,25], different strategies, such as supplying exogenous 5-ALA, introducing iron chelators, knocking down ferrochelatase (FECH), and inhibiting heme oxygenase-1 (HO-1) activity, have been investigated in an attempt to upregulate PpIX biosynthesis[22,24,26,27]. However, a rapid increase in PpIX level triggers the enhanced outflow of intratumoral PpIX for excess heme synthesis, resulting in the negative feedback of heme synthesis and downregulation of PpIX-related synthetases[3,7]. This effect renders traditional 5-ALA-PDT unable to eliminate malignant tumors because of an inadequate PpIX concentration and a short treatment time window. Inhibiting the RAS-RAF-MEK-ERK pathway could reduce FECH activity by downregulating hypoxia-inducible factor (HIF)-1α to enhance PpIX accumulation[28,29]. Hence, the continuous supply of oxygen ($O_2$) is expected to realize efficient 5-ALA-PDT by overcoming tumor hypoxia and the biosynthesis deficiency of PpIX simultaneously[30]. However, the downregulation of FECH by HIF-1α inhibitors is a slow process (~20 h), while the metabolism of 5-ALA induced PpIX in vivo is very fast (reaching METP in ~6 h and quickly clearing within 12 h). Thus, effective delivery strategies are needed to prolong the PpIX tumor retention time for efficient 5-ALA-PDT[3,8,28].

In addition, tumor heterogeneity and the complexity of applying laser-mediated drug-device combinations in clinical settings prevent the development of an ideal therapeutic strategy that ensures the best dosage of medication and laser irradiation parameters for 5-ALA-PDT[4–6]. Hence, the development of precision theranostic platforms to address the clinical limitations of 5-ALA-PDT is highly desired to improve therapeutic outcomes[31–33]. For diagnosis, it is necessary to assess the concentrations of PpIX and $O_2$ in tumor tissues in real-time via noninvasive dynamic molecular imaging and thus provides guidance about laser parameters of 5-ALA-PDT, achieving enhanced antitumor efficacy with minimal toxicity[19,34,35]. For treatment, ion-interference therapy (IIT), which uses bioactive ions (such as $Cu^{2+}$ for

glutathione (GSH) depletion and $Ca^{2+}$ for oxidative stress-mediated apoptosis) to reverse the distribution state of ions in cancer cells by changing the physicochemical properties and biological functions at the cellular level[36–38], is effectively used for regulating oxidative stress-mediated apoptosis pathways. Therefore, the combination of real-time monitoring, PDT, and IIT is promising for efficient antitumor therapy[39,40].

In this work, we develop a synthetic biology-instructed theranostic microneedle (MN) patch (denoted as MN-CCPCA) consisting of 5-ALA and catalase (CAT) co-loaded acidity-responsive copper-doped calcium phosphate nanoparticles (CCPCA NPs) and hyaluronic acid (HA). This transdermal MN patch by simultaneously delivering 5-ALA and CAT provides a synthetic biological approach to make a halt to break the negative feedback on 5-ALA-induced PpIX biosynthesis through the cascaded downregulation of HIF-1α and FECH, thus increasing the physiological concentration of PpIX by upregulation of ALA-synthetase (ALAS) and extending its METP inside tumors (Fig. 1). Such improvements in the biosynthetic kinetics of PpIX not only ensure tumor oxyhemoglobin saturation ($sO_2$) and in vivo PpIX metabolic kinetics to be monitored through fluorescence (FL)/photoacoustic (PA) duplex real-time imaging, but also provide the selectivity of irradiation parameters by allowing multiple irradiations under the single topical administration of the MN-CCPCA patch. The simultaneous supply of sufficient $O_2$ and PpIX significantly enhance the therapeutic effect of 5-ALA-PDT, while after repetitive 635 nm laser exposure, the singlet oxygen ($^1O_2$) generation by PpIX along with the released $Cu^{2+}/Ca^{2+}$ from CCPCA NPs not only consume antioxidant GSH but also achieve stronger cytosolic free $Ca^{2+}$ ($[Ca]_{CYT}$) overload for ion-interference enhanced repeatable 5-ALA-PDT[36,37,41]. These effects allow us to optimize the treatment parameters of 5-ALA-PDT in heterogeneous solid tumors (breast cancer, glioblastoma, melanoma) by the traceable FL/PA duplex imaging, and thus opens avenues for companion theranostics of potential indications of 5-ALA-PDT.

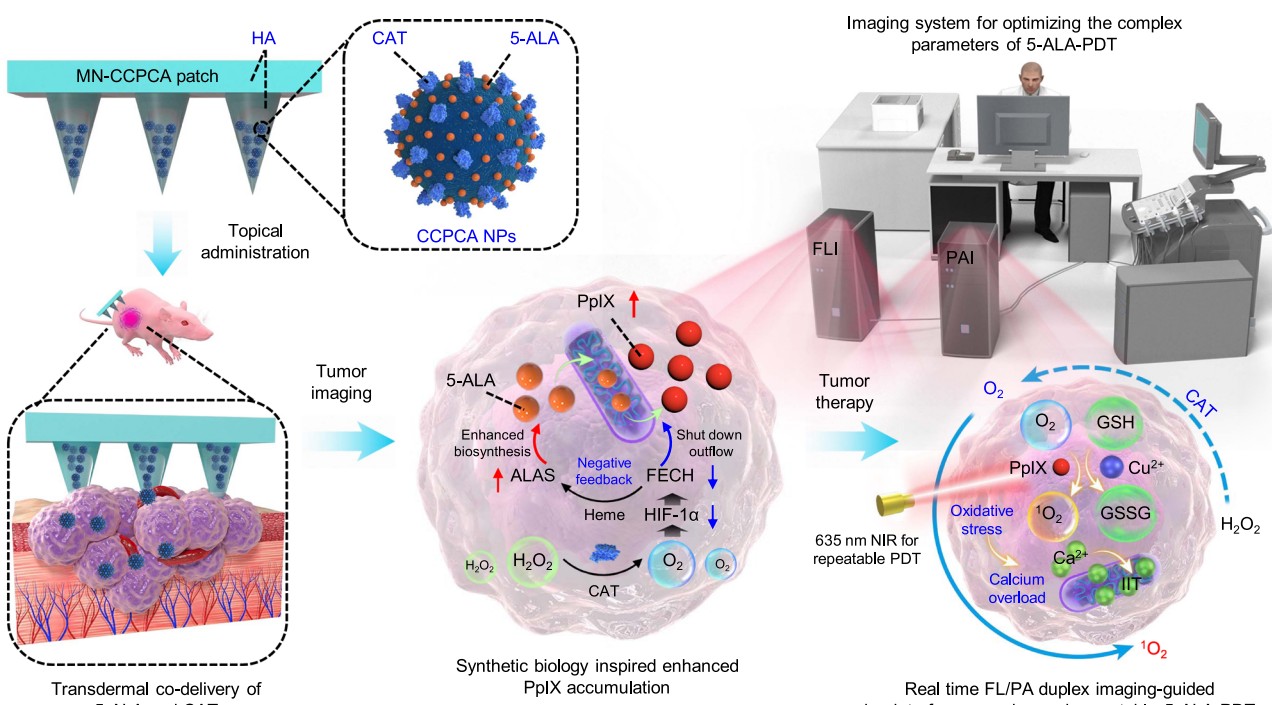

**Fig. 1 | In vivo real-time companion theranostics by MN-CCPCA patch.** 5-ALA and CAT were directly delivered to the tumor site by the transdermal co-delivery patch. The MN-CCPCA patch continuously catalyzed hydrogen peroxide ($H_2O_2$) into $O_2$ to elevate the intratumoral $sO_2$ level, which enhanced the physiological synthetic concentration and extended the METP of PpIX by "shutting down" PpIX outflow (downregulating HIF-1α and FECH) and upregulating PpIX biosynthesis (providing exogenous 5-ALA and upregulating ALAS). The platform of MN-CCPCA patch offers a strategy to monitor in vivo real-time PpIX and $sO_2$ levels and optimize treatment parameters through duplex FL and PA imaging and ultimately improves therapeutic efficacy and biosafety of 5-ALA-PDT.

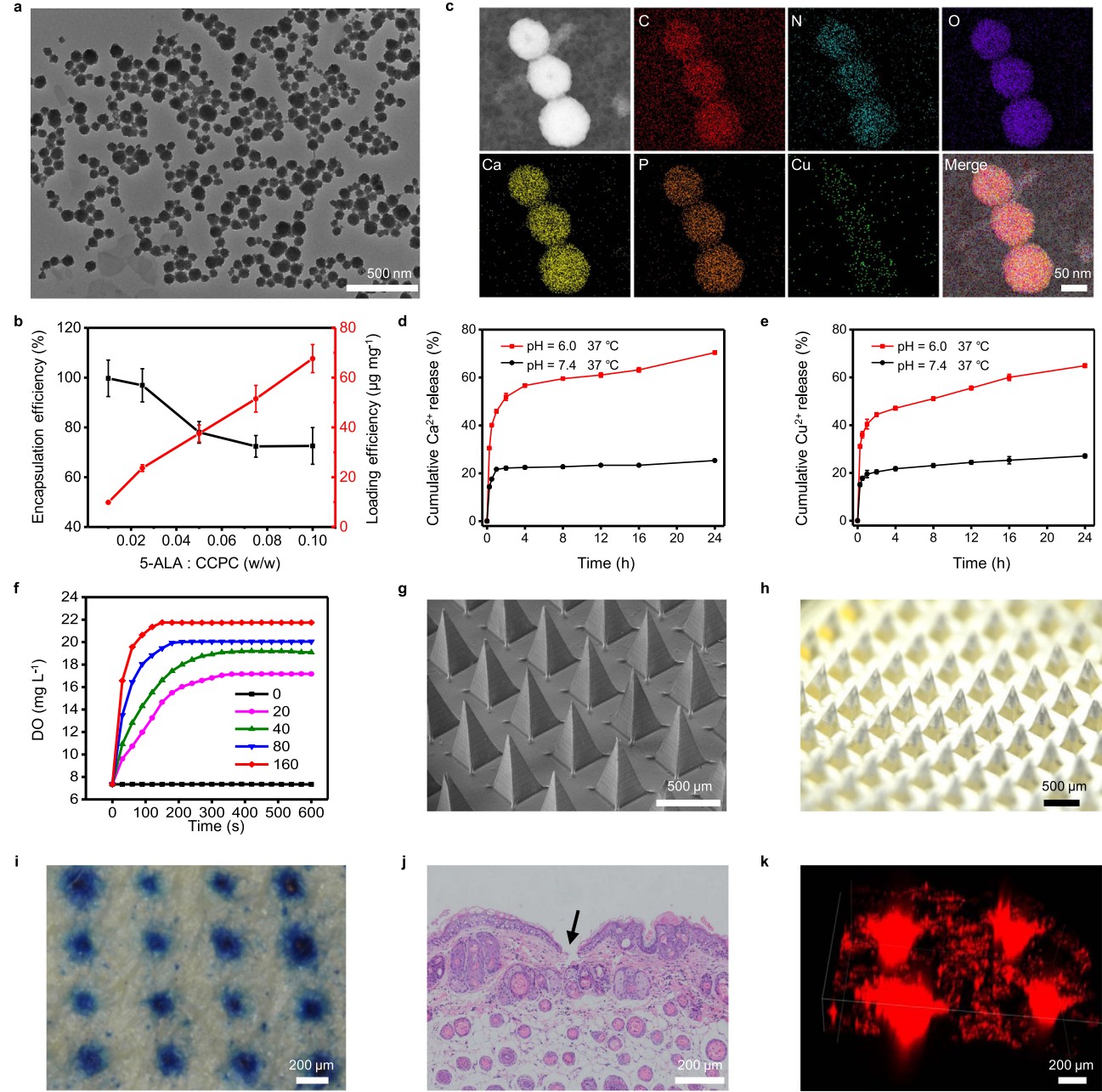

**Fig. 2 | Characterization of CCPCA NPs and MN-CCPCA patches.**
**a** Representative TEM image of CCPCA NPs, scale bar = 500 nm, $n = 3$ independent experiments. **b** Loading efficiency and encapsulation efficiency of 5-ALA at different 5-ALA: CCPC NPs feeding ratios. Data represent the means ± SD ($n = 3$). **c** Elemental mapping analysis of CCPCA NPs, scale bar = 50 nm. Images are from three independent samples of CCPCA NPs. **d** The $Ca^{2+}$ and **e** $Cu^{2+}$ release properties of CCPCA NPs in phosphate buffer solution at pH 6.0 and 7.4. Data represent the means ± SD ($n = 3$). **f** Time-dependent changes in dissolved oxygen (DO) in $H_2O_2$ (1 mM) after adding CCPCA NPs at various concentrations (0–160 μg mL$^{-1}$), $n = 3$ independent experiments. **g** Representative SEM and **h** stereo-microscopic images of the MN-CCPCA patch, scale bar = 500 μm. $n = 3$ independent experiments. **i** Representative MB staining images and **j** H&E staining images of mouse skin after the MN-CCPCA patch were applied. Images are from three biologically independent skin samples of 4-week-old female BALB/c-nude mice, $n = 3$, scale bar = 200 μm. **k** 3D reconstructed images of RhB FL on tumor skin after application of CCPCA-RhB NPs-loaded MN patches. Images are from three biologically independent skin samples of 4-week-old female BALB/c-nude mice, $n = 3$, scale bar = 200 μm. Source data are provided as a Source data file.

## Results

### Preparation and characterization of CCPCA NPs and MN-CCPCA patches

We aimed to construct a 5-ALA and CAT co-loaded delivery platform. First of all, acidity-responsive copper-doped calcium phosphate nanoparticles (CCP NPs) were obtained by a modified biomineralization method using bovine serum albumin (BSA) as a biomimetic template[12,42]. CAT was then loaded via an amidation reaction, followed by the loading of 5-ALA under magnetic stirring at room temperature, resulting in CCPCA NPs. Morphological characterization by transmission electron microscopy (TEM, Fig. 2a) and scanning electron microscopy (SEM, Supplementary Fig. 1a) indicated the spherical shape of both CCP NPs (Supplementary Fig. 1b) and CCPCA NPs with good monodispersity. Thermogravimetric analysis confirmed 7.9 wt% CAT in CCPCA NPs (Supplementary Fig. 1c). We also observed a linear relationship between the 5-ALA loading efficiency and the feeding ratio

of 5-ALA: CAT-loaded CCP (CCPC) NPs (Fig. 2b and Supplementary Fig. 1d). For all subsequent experiments, we chose the maximum loading efficiency of 6.8% for 5-ALA.

After co-loading CAT and 5-ALA, we found that CCPCA NPs had a less negative surface charge with a slight increase in hydrodynamic size than CCP NPs (Supplementary Fig. 1e–g). TEM elemental mapping images suggested that CCPCA NPs were composed of C, N, O, Cu, Ca, and P elements (Fig. 2c), while inductively coupled plasma atomic emission spectrometer (ICP-AES) reported 6.3% Cu and 29.7% Ca in the CCPCA NPs. Notably, both $Ca^{2+}$ (70.4%) and $Cu^{2+}$ (64.9%) rapidly released from CCPCA NPs under acidic (pH 6.0) than neutral (pH 7.4) conditions (25.3% and 27.1%, respectively), indicating pH-dependent ion release (Fig. 2d, e). Morphological characterization of CCPCA NPs by TEM also revealed that the CCPCA NPs completely degraded into tiny fragments at pH 6.0, confirming the pH-dependent behavior of CCPCA NPs (Supplementary Fig. 1h, i).

Since CAT was loaded by CCPCA NPs, we next tested the catalytic ability of CCPCA NPs. CAT alone and CCPCA NPs showed a similar time-dependent catalytic decomposition of $H_2O_2$, suggesting that CCPCA NPs did not disturb the catalytic activity of CAT (Supplementary Fig. 2a–e). In addition, the generation of $O_2$ by CCPCA NPs was proportional to the concentration of CCPCA NPs and $H_2O_2$ (Fig. 2f and Supplementary Fig. 2f). These findings confirmed the successful co-loading of CAT and 5-ALA by CCPCA NPs.

To increase the bioavailability of 5-ALA, we further integrated CCPCA NPs into a HA-based MN patch. MNs loaded with different NPs, such as CCP, CCPC, 5-ALA-loaded CCP (CCPA), and CCPCA NPs, were prepared through a two-layer strategy by applying a poly-dimethylsiloxane (PDMS) mold (Supplementary Fig. 3). SEM (Fig. 2g and Supplementary Fig. 4a) and optical stereomicroscope images (Fig. 2h and Supplementary Fig. 4b) showed that the MN-CCPCA patch had pyramid-shaped needles with a sharp tip and uniform size. Compared to the smooth morphology of a naked tip, the high-resolution SEM image showed an irregular and rough surface of the CCPCA NPs-loaded tip (Supplementary Fig. 4c, d).

Notably, the MN-CCPCA patch displayed good mechanical strength of 0.39 N/needle destructive force (Supplementary Fig. 5a), allowing the MN-CCPCA patch to penetrate the skin effectively[43,44]. Meanwhile, both methylene blue (MB, Fig. 2i and Supplementary Fig. 5b) and hematoxylin and eosin (H&E, Fig. 2j and Supplementary Fig. 5c) staining images of mice and rats indicated that the tips of the MN-CCPCA patch formed microchannels in the skin. We also loaded rhodamine B (RhB) onto CCPCA (CCPCA-RhB) NPs to simulate the infiltration and distribution of CAT and 5-ALA (Supplementary Fig. 5d). The three-dimensional (3D) reconstructed RhB FL imaging of MN patch (loaded with CCPCA-RhB NPs)-treated skin of A375 melanoma tumors also identified several micropores in the skin (Fig. 2k), confirming the skin permeability of the MN-CCPCA patch to deliver both 5-ALA and CAT locally at the target site.

Together, these results indicate that CCPCA NPs can be successfully loaded into the MN patch. More importantly, the MN-CCPCA patch has sufficient strength to cross the epidermal barrier of solid tumors, thereby selectively and painlessly delivering CCPCA NPs and improving the bioavailability of 5-ALA and CAT.

## PpIX biosynthesis and outflow "shut down"
The enzymes required for PpIX biosynthesis compete with respiration complexes for molecular oxygen, while the Warburg effect disturbs the PpIX and $Fe^{2+}$ balance in the mitochondria of cancer cells, leading to a higher accumulation of PpIX in cancer cells than in normal cells[7,25,27]. As both 5-ALA and CAT are co-loaded into CCPCA NPs, we hypothesized that continuous $O_2$ generation by catalytic endogenous $H_2O_2$ decomposition will alter the physiological environment of PpIX biosynthesis and alleviate tumor hypoxia and mediate the cascaded downregulation of HIF-1α and FECH to ultimately shutdown the

outflow door of PpIX. By this reasoning, ALAS will also be upregulated by a false negative feedback signal of "heme deficiency", leading to enhanced PpIX accumulation because of upregulated endogenous and exogenous PpIX biosynthesis processes (Fig. 3a).

To prove our hypothesis, we studied intracellular $O_2$ generation using a tris(4,7-diphenyl-1,10-phenanthroline) ruthenium (II) dichloride [Ru (DPP)$_3$] Cl$_2$ (RDPP) probe. Compared to the CCP and CCPA groups, a significant decrease in the RDPP FL signal was observed in both the CAT-loaded CCPC (29.1%) and CCPCA (28.6%) groups (Fig. 3b, c). Moreover, the lower relative intracellular $H_2O_2$ content in the CCPC (30.6%) and CCPCA (27.4%) groups also demonstrated the in vitro catalytic activity of CAT in human A375 melanoma cells (Fig. 3d), suggesting sufficient in situ $O_2$ generation ability to alleviate tumor hypoxia.

To assess the ability of CCPCA NPs to enhance PpIX production, we incubated 5-ALA and different NPs (CCP, CCPC, CCPA, CCPCA) with A375 cells to monitor the in vitro production of PpIX in real time. Compared to non-5-ALA-loaded NPs (CCP and CCPC), the relative FL intensities of PpIX were much higher in 5-ALA (139.8%), CCPA (166.2%), and CCPCA NPs (225.0%) at 24 h post-incubation (Fig. 3e, f). Meanwhile, these groups also displayed 5-ALA concentration-dependent increase in the PpIX FL intensity (Supplementary Fig. 6a, b), indicating that the exogenous administration of high doses of 5-ALA specifically enriches PpIX accumulation in A375 cells[21–23]. It is worth noting that the amount of intracellular PpIX at 24 h post-incubation with CCPCA ($141.4 \pm 1.7$ ng/$10^6$ cells) was ~1.9- and 4.0-fold higher than that of CCPA ($72.4 \pm 7.7$ ng/$10^6$ cells) and 5-ALA ($35.1 \pm 3.3$ ng/$10^6$ cells, 10.8 μg mL$^{-1}$) at the same amount of 5-ALA (Fig. 3g and Supplementary Fig. 6c). The real-time PpIX metabolic curves indicated the maximum FL intensities of PpIX in the 5-ALA and CCPA groups at 6 and 10 h post-incubation (Fig. 3h and Supplementary Fig. 6d–e), respectively, and then gradually decreased over time. By contrast, the CCPCA group exhibited the maximum FL intensity of PpIX at 24 h post-incubation, suggesting that the CAT and 5-ALA co-delivery platform could extend the METP of PpIX, thus providing a longer theranostic window for repeatable 5-ALA-PDT.

The FL intensity of PpIX at METP in the CCPCA group (24 h) was also 2.5 and 2.9 times that of the CCPA (10 h) and 5-ALA (6 h) groups, respectively (Fig. 3h). This result indicated enhanced PpIX accumulation, likely because of an imbalance between PpIX synthesis and its outflow in the CCPCA-treated A375 cells. Importantly, the continuous elevation of endogenous $O_2$ levels by CAT suppressed the expression of HIF-1α in both normoxia (21% $O_2$) and hypoxia conditions (1% $O_2$, Fig. 3i–k). The decreased expression of HIF-1α was observed in a time-dependent (Fig. 3l) as well as CCPCA concentration-dependent (Fig. 3m) manner, which further reduced the expression of FECH (Fig. 3i), thereby restricting the outflow of PpIX and triggering the negative feedback process of heme shortage to increase ALAS expression (Fig. 3i). As a result, the exogenous 5-ALA delivered by CCPCA NPs and the endogenous synthetic 5-ALA by cancer cells could be simultaneously converted into PpIX, further increasing the concentration of synthetic PpIX[7,24,28]. These results demonstrate that PpIX enrichment in cancer cells can be biologically upregulated following 5-ALA and CAT co-delivery.

## Synergistic mechanism of ion interference-enhanced 5-ALA-PDT
Low dark toxicity and high phototoxicity are necessary to improve the prognosis of PDT in the clinic[4–6]. Due to acidity-responsive degradation properties, CCPCA NPs (160 μg mL$^{-1}$)-treated human embryonic kidney 293T cells and mouse breast cancer 4T1 cells showed 94.6% (Fig. 4a) and 79.6% (Fig. 4b) viabilities in the dark after 24 h of incubation respectively, suggesting that CCPCA NPs have better biocompatibility with normal cells than cancer cells. Meanwhile, we also found concentration-dependent phototoxicity of CCPCA NPs after laser irradiation, as indicated by more than 80% cellular death of both

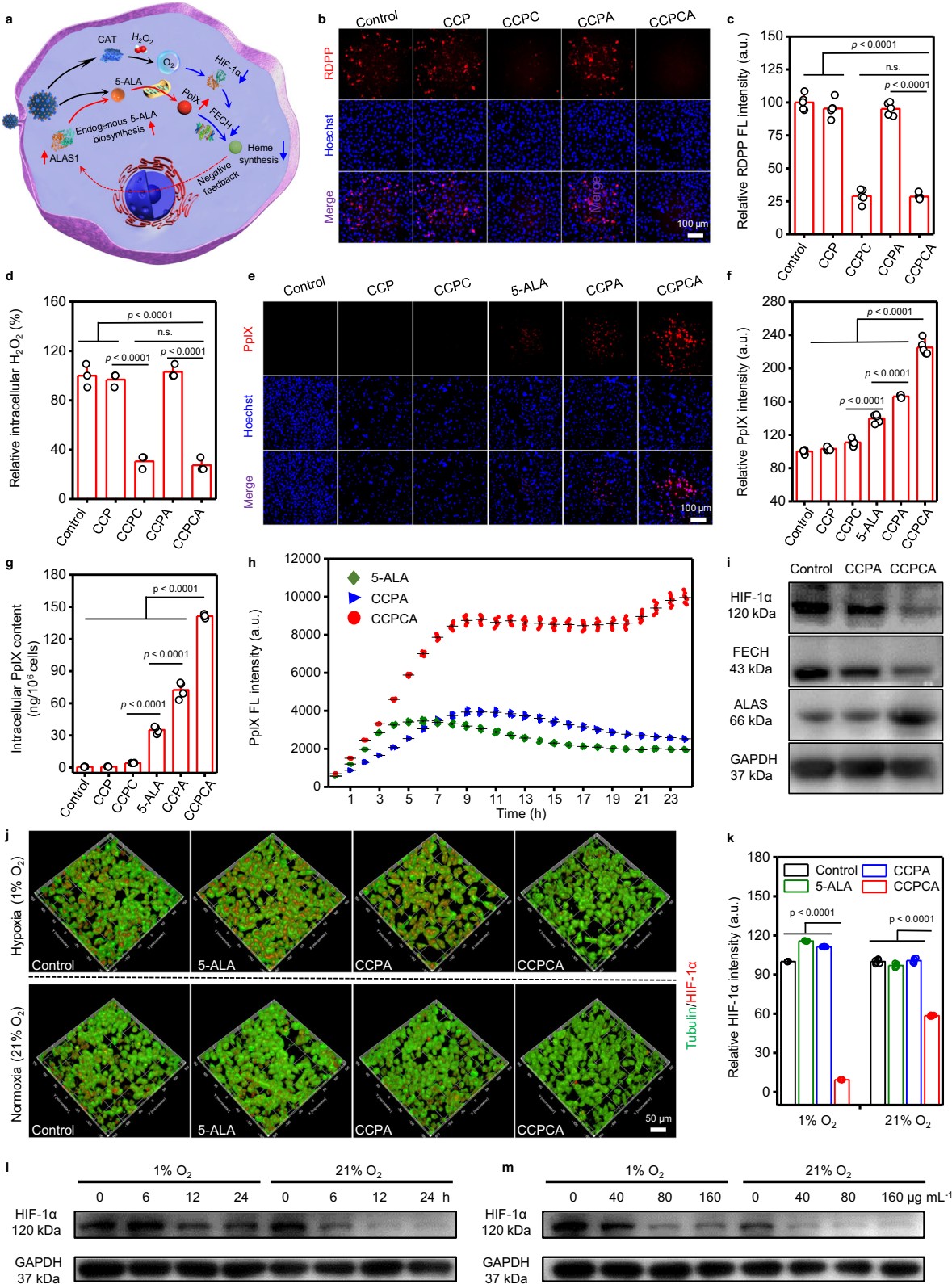

A375 and malignant glioblastoma U87MG cells (Fig. 4c, d). By contrast, CCPA NPs-treated A375 and U87MG cells showed improved viabilities under the same laser irradiation conditions and 5-ALA dosage. Flow cytometric analysis of cell apoptosis further confirmed the findings of dark cytotoxicity and phototoxicity assays, combined with ~2–3-fold higher levels of apoptotic A375 cells after CCPCA + Laser (+) than 5-ALA (+) and CCPA (+) treatment (Fig. 4e) under normoxia

condition. Similarly, after irradiation under hypoxia condition for 5 min, a significant enhancement of the FL intensity of caspase-3 in the CCPCA group (193.6%) was observed compared with the 5-ALA (111.3%) and CCPA (151.4%) groups (Fig. 4f, g), while the expression of HIF-1α in the CCPCA (+) group was only 26.4% of the control (+) group (Fig. 4h), indicating that CCPCA can improve the phototoxicity both in normoxia and hypoxia conditions.

**Fig. 3 | 5-ALA and CAT co-delivery by CCPCA NPs upregulates PpIX accumulation in vitro. a** Increased intracellular $O_2$ induced PpIX accumulation through a heme negative feedback mechanism. **b** FL images of RDPP and **c** corresponding quantitative analysis after incubation with different NPs (CCP, CCPC, CCPA, CCPCA) at equivalent concentrations (80 μg mL$^{-1}$) for 24 h, scale bar = 100 μm. Images are representative of five independent replicates. Data represent the means ± SD ($n = 5$). Statistical significance was calculated via one-way ANOVA with Tukey's multiple comparisons test. **d** Quantitative analysis of $H_2O_2$ levels in A375 cells after viaious treatments. Statistical significance was calculated via one-way ANOVA with Tukey's multiple comparisons test. $n = 3$ independent experiments. **e** FL images and **f** corresponding quantitative analysis of PpIX in A375 cells after exposure to equivalent amounts of free 5-ALA (5.4 μg mL$^{-1}$) or 80 μg mL$^{-1}$ NPs (CCP, CCPC, CCPA, CCPCA), scale bar = 100 μm. Images are representative of five independent replicates. Data represent the means ± SD ($n = 5$). Statistical significance was calculated via one-way ANOVA with Tukey's multiple comparisons test. **g** Intracellular PpIX content in the lysate after 24 h incubation with free 5-ALA (10.8 μg mL$^{-1}$) or NPs (160 μg mL$^{-1}$). Statistical significance was calculated via one-way ANOVA with Tukey's multiple comparisons test. $n = 4$ independent experiments. **h** Real-time FL quantitative analysis of PpIX metabolism after incubation with free 5-ALA (10.8 μg mL$^{-1}$) or NPs (160 μg mL$^{-1}$). Data represent the means ± SD ($n = 7$). Statistical significance was calculated via one-way ANOVA with Tukey's multiple comparisons test. **i** HIF-1α, FECH, and ALAS expression in A375 cells determined by western blotting. $n = 3$ independent experiments. **j** Immunofluorescence images and **k** quantitative FL intensity analysis of anti-α-tubulin (green) and anti-HIF-1α (red) in A375 cells after different treatments under hypoxic (1% $O_2$, 5% $CO_2$, and 94% $N_2$) and normoxic (21% $O_2$, 5% $CO_2$, and 74% $N_2$) conditions, respectively. Scale bar = 50 μm. Images are representative of six independent replicates. Data represent the means ± SD ($n = 6$). Statistical significance was calculated via two-tailed Student's $t$ test. Western immunoblots analysis of HIF-1α expression levels in A375 cells treated with CCPCA NPs at different **l** time (treated with 80 μg mL$^{-1}$ CCPCA NPs) and **m** concentrations (treated for 24 h). $n = 3$ independent experiments. Source data are provided as a Source data file.

To determine the synergistic mechanism of IIT and 5-ALA-PDT, we monitored the physiological functions of A375 cells before and after laser irradiation. Resting [Ca]$_{CYT}$ is meticulously controlled in the cells as [Ca]$_{CYT}$ overload is a cell death-promoting factor. [Ca]$_{CYT}$ overload can be logically achieved through the combination of oxidative stress produced by $^1O_2$ and exogenous $Ca^{2+}$ influx[36,37,41,45]. As exogenous 5-ALA was provided and converted to PpIX, both CCPA and CCPCA NPs induced strong [Ca]$_{CYT}$ overload under laser irradiation; However, the CCPCA (+) group exhibited an increased FL intensity of [Ca]$_{CYT}$ by 19.5% compared with that of the CCPA (+) group due to CAT loading (Supplementary Fig. 7a, b). Similarly, the FL intensity of [Ca]$_{CYT}$ recorded in the CCPCA (+) group was 1.7-fold higher than that of the 5-ALA (+) treatment group (Supplementary Fig. 7a, b). These results are attributed to the fact that the CCPCA NPs could promote the accumulation of higher concentrations of PpIX and $O_2$ in A375 cells (Fig. 3b–h), thus simultaneously increasing the $^1O_2$ level under laser irradiation and reducing the GSH level (Fig. 4i) by releasing $Cu^{2+}$ to amplify oxidative stress-mediated [Ca]$_{CYT}$ overload, achieving ion interference-enhanced 5-ALA-PDT.

We next evaluated the oxidative stress level of A375 cells using a 2,7-dichlorodihydrofluorescein diacetate (DCFH-DA) assay. Under laser irradiation, we noted a significant enhancement in the FL intensity of oxidized intracellular DCF in the CCPCA group (189.9%) compared with the 5-ALA (113.9%) and CCPA (146.4%) groups (Fig. 4j, k), suggesting the synergistic effect between intracellular $^1O_2$ and $Ca^{2+}$/$Cu^{2+}$ interferences. By contrast, the reactive oxygen species (ROS) levels in nonirradiated groups were similar to those of the control group (Supplementary Fig. 8a, b), verifying that laser irradiation is essential for inducing the oxidative stress needed to ensure a synergistic IIT/PDT effect.

Mitochondrial membrane potential (ΔΨm) is an important indicator of mitochondrial function and is used to assess the early period of cell apoptosis[45,46]. After different treatments, we analyzed the changes in ΔΨm by 5,5,6,6-tetrachloro-1,1,3,3-tetraethyl-imidacarbocyanine JC-1 assay. We saw that the FL intensity ratio of JC-1 aggregates/JC-1 monomer in the CCPCA group (1.3 ± 0.1) was significantly reduced compared with the CCPA (2.4 ± 0.1) and 5-ALA (4.8 ± 0.3) groups under laser irradiation (Fig. 4l, m). By contrast, we saw no noticeable decrease in any other groups in the dark (Supplementary Fig. 8c, d), indicating that $^1O_2$ generation under laser irradiation can reduce ΔΨm, which is closely related to the levels of intracellular [Ca]$_{CYT}$, PpIX and $O_2$.

Together, CCPCA NPs could provide sufficient $O_2$ and PpIX for 5-ALA-PDT and consume GSH through $Cu^{2+}$ to increase the lethality of $^1O_2$. The resulting oxidative stress promoted ΔΨm damage and insufficient $Ca^{2+}$ buffering capacity, leading to [Ca]$_{CYT}$ overload and cell apoptosis (Fig. 4n). Remarkably, 5-ALA-PDT alone and IIT in the dark exhibited poor therapeutic efficacy, while under laser irradiation, the synergistic effect of $^1O_2$ and exogenous $Ca^{2+}$/$Cu^{2+}$ interferences by CCPCA NPs realized enhanced therapeutic efficacy through an apoptosis-promoting pathway.

## Monitoring PpIX biosynthesis in vivo by real-time FL/PA duplex imaging

As oxidative stress is a key to the synergistic effect of IIT and 5-ALA-PDT for enhanced therapeutic efficacy[37,47,48], we explored the metabolic dynamics of PpIX and $O_2$ generation to validate the feasibility for PpIX synthesis regulation and optimizing treatment parameters in vivo. We applied different MN patches (I: Control; II: MN-CCP; III: MN-CCPC; IV: MN-CCPA; V: MN-CCPCA) onto A375 tumor-bearing mice, and then conducted in vivo FL imaging of PpIX and PA imaging of intratumoral sO$_2$ levels. The MN-CCPA group exhibited a gradual increase in the PpIX FL signal at the tumor site until 12 h (Fig. 5a, b). By contrast, there was no detectable FL signal in the MN-CCP group (Fig. 5a, b), suggesting that the local delivery of exogenous 5-ALA could significantly enhance intratumoral PpIX accumulation. Notably, the PpIX FL intensity obtained by MN-CCPCA treatment (12 h) was 2.2-fold higher than that obtained by MN-CCPA (12 h) treatment at the METP of PpIX (Fig. 5b), indicating that 5-ALA and CAT co-delivery could promote PpIX tumor accumulation in vivo. This result is also in accordance with our in vitro findings (Fig. 3e–h).

The FL cross-section of PpIX at the tumor site also indicated that all MN patches could effectively penetrate the stratum corneum to reach deep tumor tissues (Fig. 5c). Compared to the control group, 2.3-fold higher sO$_2$ level in tumors was observed in the MN-CCPCA group at 24 h post-treatment (Fig. 5d), confirming the alleviation of the hypoxic tumor microenvironment in vivo. However, the MN-CCPC group still presented a higher sO$_2$ level at 48 h (24 h post-irradiation). This higher sO$_2$ level is ascribed to the lack of PpIX biosynthesis after MN-CCPC treatment, thus resulting in a higher sO$_2$ level after laser irradiation than that of MN-CCPCA treatment (Fig. 5d). Even so, the level of sO$_2$ in the MN-CCPCA group was 1.7 and 2.5 times than that of the control and MN-CCPA groups at 24 h post-irradiation, suggesting that the MN-CCPCA patch could relieve tumor hypoxia during intermittent irradiation (Fig. 5e).

These findings demonstrate that intratumoral PpIX accumulation can be upregulated in vivo by MN-CCPCA patch. The in vivo real-time metabolism quantitative analysis of PpIX and sO$_2$ by FL/PA duplex imaging also creates the necessary conditions to optimize laser irradiation parameters for repeated 5-ALA-PDT.

## Improved in vivo 5-ALA PDT efficacy by CCPCA

Inspired by the in vitro results, we next developed an A375 xenograft tumor model to evaluate the antitumor effects of MN-CCPCA patch for ion interference-enhanced repeatable 5-ALA-PDT. When tumor sizes reached ~70 mm$^3$, we randomly divided the A375 tumor-bearing mice into six groups: (I) MN, (II) MN-CCP, (III) MN-CCPC, (IV) MN-CCPCA, (V)

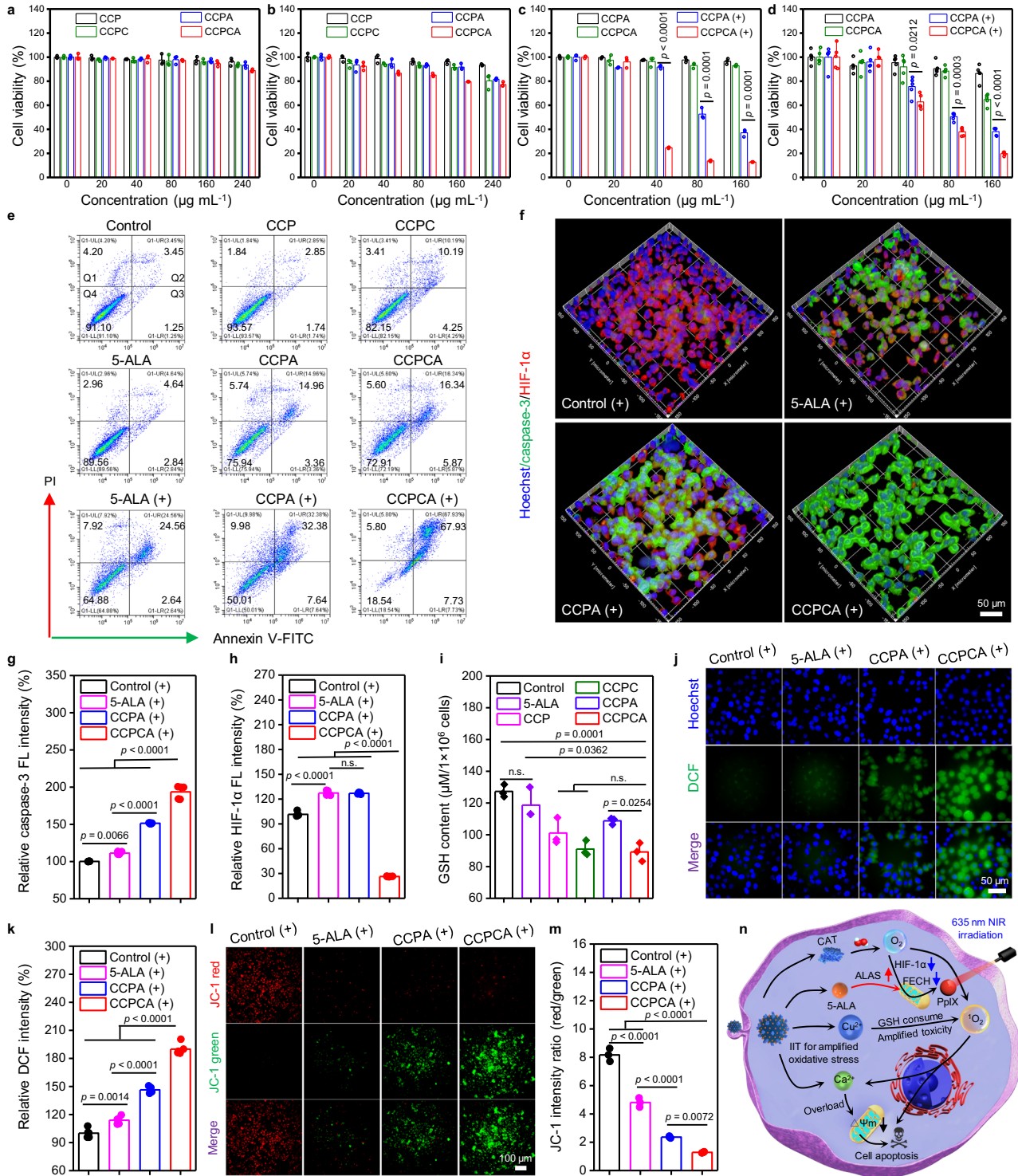

MN-CCPA + Laser (+), and (VI) MN-CCPCA (+). For the laser irradiation groups, the mice were exposed to a 635 nm laser at 6, 12, and 24 h post-transdermal application of patches under the navigation of FL/PA duplex imaging (Fig. 6a). Under laser irradiation, the MN-CCPCA (+) group showed effective tumor suppression with a tumor inhibition rate of 91.2%, which was significantly higher than that seen in the MN-CCPA (+, 34.9%) and MN-CCPCA (31.6%) groups (Fig. 6b, c). We ascribed such substantial tumor inhibition to the synergistic antitumor effects between IIT and 5-ALA-PDT. Importantly, although the mice in the MN-CCPA (+) and MN-CCPCA groups showed significant tumor inhibition within 1 week, the tumors relapsed completely after 2 weeks

(Fig. 6b, c), suggesting that inadequate 5-ALA-PDT (short supply of PpIX or $O_2$) combined IIT or IIT alone is unable to eradicate tumors without any recurrence. By contrast, mice in the MN-CCPCA (+) group did not show any tumor recurrence within two weeks (Fig. 6d, e and Supplementary Fig. 9a).

We next resected the tumors after different treatments to evaluate their proliferative capacity and verify the synergistic antitumor effect. Immunofluorescence staining images of HIF-1α showed reduced expression in both the MN-CCPC and MN-CCPCA groups compared to non-CAT-loaded groups (Fig. 6f). It is worth noting that the FL intensity of HIF-1α in the MN-CCPA (+) group after three

**Fig. 4 | Synergistic therapeutic mechanism of IIT and 5-ALA-PDT.** Cytotoxicity of different NPs in **a** 293T cells and **b** 4T1 cells after co-incubation with CCP/CCPC/CCPA/CCPCA NPs for 24 h. The dark toxicity and phototoxicity of CCPA and CCPCA NPs with **c** A375 cells and **d** U87MG cells. Data represent the means ± SD (**a**–**c**, $n = 3$, **d**, $n = 5$). Statistical significance was calculated via two-tailed Student's $t$ test. **e** Annexin V-FITC/PI apoptosis analyses (FACS gating strategy) of A375 cells after the indicated treatments. Images are representative of three biologically independent replicates ($n = 3$). Q1, dead cells; Q2, late apoptotic cells; Q3, early apoptotic cells; Q4, live cells. The symbols "+" indicate laser irradiation. **f** Immunofluorescence images and corresponding quantitative FL intensity analysis of **g** anti-caspase-3 (green) and **h** anti-HIF-1α (red) in A375 cells after different treatments under hypoxic (1% $O_2$, 5% $CO_2$, and 94% $N_2$) conditions. A375 cells were incubated with blank DMEM, 5-ALA, CCPA, and CCPCA NPs for 24 h and then irradiation with a 635 nm laser (200 mW $cm^{-2}$, 5 min), scale bar = 50 μm. Images are

representative of five independent replicates. Data represent the means ± SD ($n = 5$). Statistical significance was calculated via one-way ANOVA with Tukey's multiple comparisons test. **i** The GSH level in A375 cells after incubation with free 5-ALA (5.4 μg $mL^{-1}$) and different NPs (80 μg $mL^{-1}$) for 24 h. Statistical significance was calculated via one-way ANOVA with Tukey's multiple comparisons test. $n = 3$ independent experiments. **j** Intracellular FL images of ROS stained with DCFH-DA and **k** corresponding quantification, **l** JC-1 and **m** corresponding quantification after treatment with blank medium, 5-ALA, CCPA and CCPCA NPs (equivalent 5.4 μg $mL^{-1}$ 5-ALA) for 24 h and then irradiation with a 635 nm laser (200 mW $cm^{-2}$, 5 min), scale bar = 50 μm. Data represent the means ± SD (**k**, $n = 5$, **m**, $n = 3$). Statistical significance was calculated via one-way ANOVA with Tukey's multiple comparisons test. **n** Synergistic mechanism of IIT and 5-ALA-PDT. Source data are provided as a Source data file.

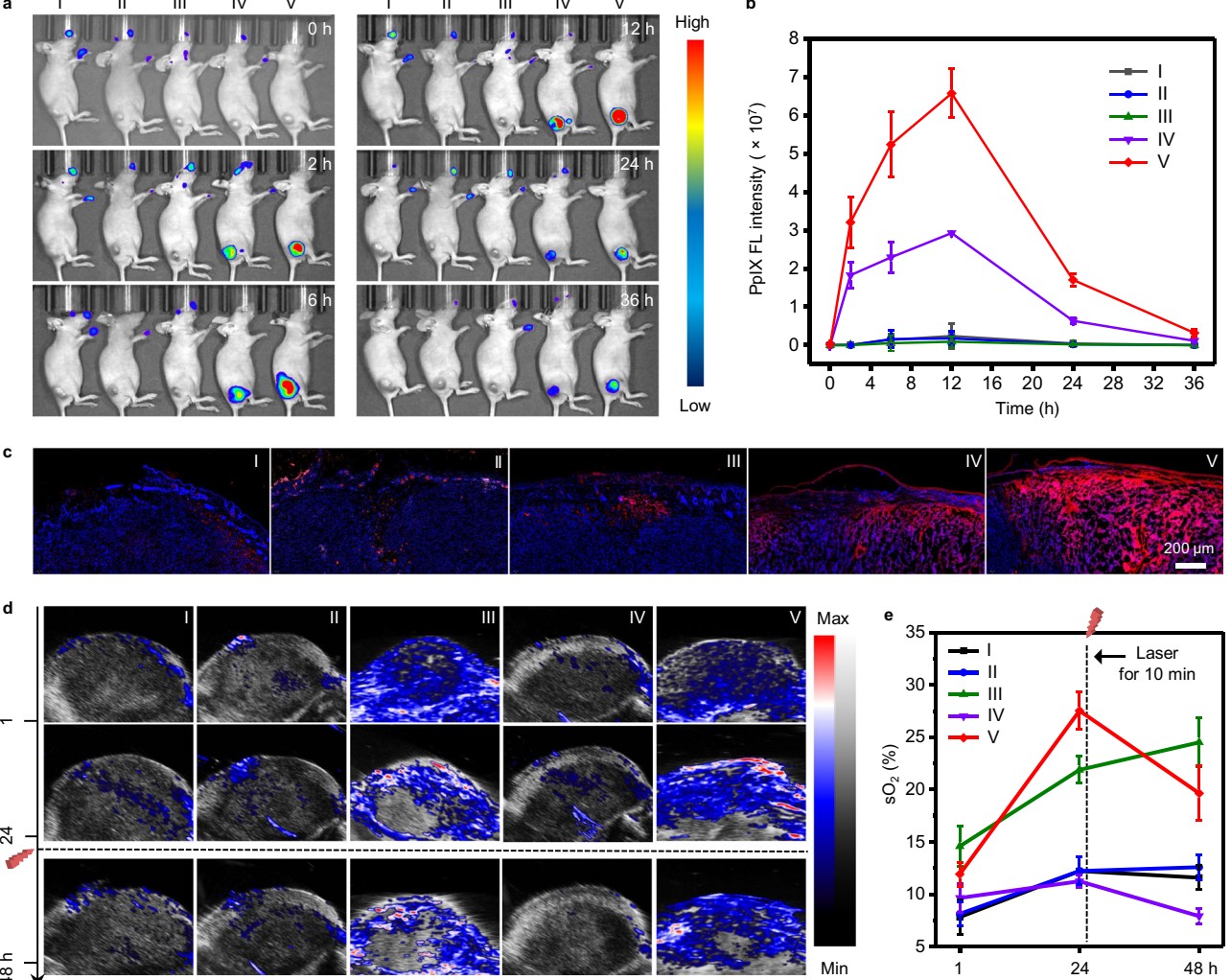

**Fig. 5 | In vivo real-time quantitative analysis of PpIX metabolism and sO₂ levels by FL/PA duplex imaging. a** In vivo PpIX FL images of A375 tumor-bearing mice visualized using an IVIS spectrum system after viaious treatments (I: Control; II: MN-CCP; III: MN-CCPC; IV: MN-CCPA; V: MN-CCPCA). **b** The FL intensity of the tumor site after administration of MN patches at the indicated time points. Data represent the means ± SD from three biologically independent female A375 tumor-bearing mice, $n = 3$. **c** Biodistributions of different MN patch-derived PpIX in skin and tumor were monitored by FL cross sections (red from PpIX and blue from

Hoechst), scale bar = 200 μm. Images are from three biologically independent tumors of 4-week-old female A375 tumor-bearing mice, $n = 3$. **d** Representative PA images of sO₂ in A375 tumors before and after viaious treatments at a given time interval. All the groups were irradiated with a 635 nm laser (200 mW $cm^{-2}$, 10 min) after testing the sO₂ at the 24 h time point. **e** Quantitative analysis of the sO₂ in the total tumor area at different time points. Data represent the means ± SD from three biologically independent female A375 tumor-bearing mice, $n = 3$. Source data are provided as a Source data file.

consecutive rounds of laser irradiation (200 mW $cm^{-2}$, 10 min each time) was 2.1-fold higher than that of the CCPCA ( + ) group (Fig. 6f and Supplementary Fig. 9b), suggesting that local 5-ALA-PDT without an $O_2$ supply would significantly increase HIF-1α expression in tumor tissues,

thereby severely limiting the therapeutic effect. By contrast, HIF-1α expression in the MN-CCPCA ( + ) group was 47.3% lower than that of the control group (Fig. 6f and Supplementary Fig. 9b), as we also seen in vitro (Fig. 4f–h) and in vivo (Fig. 5d, e).

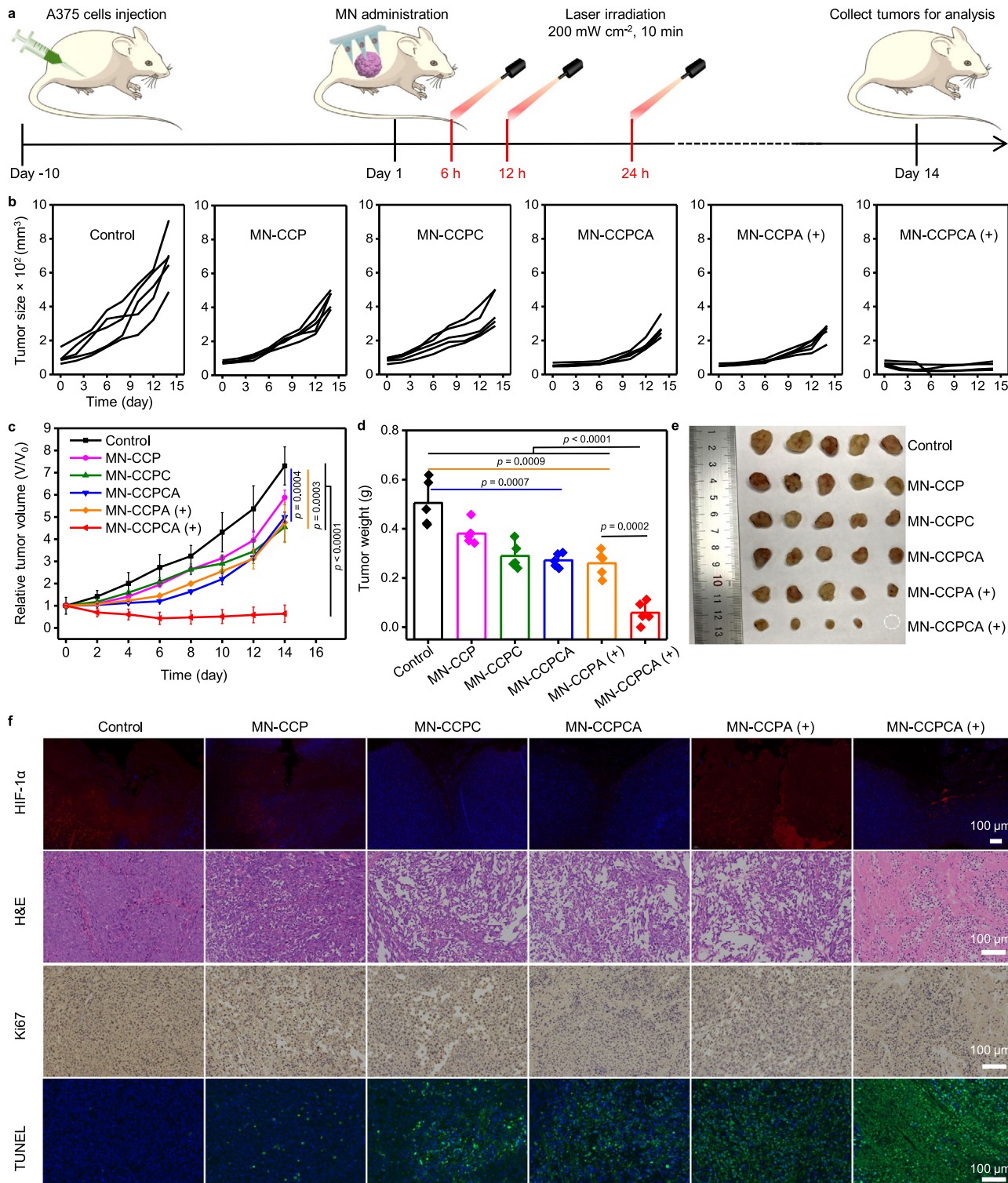

**Fig. 6 | The MN-CCPCA patch inhibits tumor recurrence by ion interference-enhanced repeatable 5-ALA-PDT. a** Schedule of the therapeutic process upon FL/PA duplex imaging navigation. **b** Individual tumor growth curves and **c** relative growth curves of A375 tumors in different treatment groups. **d** Average weight of A375 tumors at the end of the treatment period. All data represent the means ± SD from five biologically independent female A375 tumor-bearing mice, $n = 5$. Statistical significance was calculated with two-tailed Student's $t$ test. **e** Photographs of A375 tumors ($n = 5$) dissected from the mice on day 14 after exposure to different treatments. **f** HIF-1α-, H&E-, Ki67-, and TUNEL-stained sections from A375 tumor-bearing mice after various treatments indicated. Images are from five biologically independent tumors of 6-week-old female BALB/c-nude mice, $n = 5$. Scale bar = 100 μm. The symbols "+" indicate laser irradiation. Source data are provided as a Source data file.

H&E staining images indicated the most severe karyopyknosis and plasmatorrhexis in MN-CCPCA (+) treated tumors compared with other groups, while Ki67 staining results indicated an obvious inhibition of tumor cell proliferation (Fig. 6f). Furthermore, TdT (terminal deoxynucleotidyl transferase)-mediated dUDP nick-end labeling (TUNEL) staining showed the most severe cellular apoptosis in the MN-CCPCA (+) group, while the apoptosis FL intensities of tumors for MN-CCPCA and MN-CCPA (+) treatment groups were 49.1% and 60.3% of the MN-CCPCA (+) group (Fig. 6f and Supplementary Fig. 9c).

Importantly, the body weight of A375 tumor-bearing mice in both the MN-CCPCA and MN-CCPCA (+) groups was slightly increased during the treatment period (Supplementary Fig. 9d), with no obvious damage to the major organs, as shown by H&E staining images (Supplementary Fig. 9e). We thus conclude that the MN-CCPCA patch has good biocompatibility for in vivo ion interference-enhanced repeatable 5-ALA-PDT.

## MN administration improved the bioavailability and therapeutic effect of 5-ALA-PDT in vivo

To confirm whether our developed theranostic microneedle patch (MN-CCPCA) could effectively treat solid malignant tumors by optimizing the delivery strategies and complex parameters of 5-ALA-PDT, the biological distributions of PpIX after intravenous injection (i.v.), intratumoral injection (i.t.) or MN administration of 5-ALA and CCPCA were determined by FL imaging, respectively. Because of the high interstitial fluid pressure of 4T1 solid tumors[49], negligible FL intensity of PpIX was seen at tumor sites after i.v. administration of 5-ALA or CCPCA (Fig. 7a, b). The METPs of intratumoral PpIX in the 5-ALA i.t., CCPCA i.t., MN-5-ALA, and MN-CCPCA groups were 1, 12, 12, and 24 h, respectively (Fig. 7a, b). The 2.3- and 4.7-fold higher FL intensities of PpIX in the MN-CCPCA group were recorded at METP (24 h) than those of CCPCA i.t. and MN-5-ALA groups, respectively. Both in vivo and ex vivo PpIX FL images of 4T1 tumor-bearing mice indicated that a higher concentration of PpIX and longer tumor retention time can be achieved in 4T1 tumors by MN-CCPCA treatment (Supplementary Fig. 10a–d).

Because intermittent irradiation during PDT can alleviate hypoxia[50,51], laser irradiation at the point when the maximum PpIX accumulates in the tumors combined with a long irradiation interval (intermittent irradiation) will facilitate the restoration of intratumoral $O_2$ levels, resulting in high $^1O_2$ generation to realize efficient antitumor effects. Based on these facts, we selected 12, 24, and 36 h as the multiple (M) irradiation time (200 mW cm$^{-2}$, 10 min each time) for the MN-CCPCA (M) group (Fig. 7c). The other treatment groups received a single (S) irradiation (200 mW cm$^{-2}$, 30 min) at the METP of PpIX according to its metabolic kinetic analysis (Fig. 7a, b). The 4T1 tumor-bearing mice were randomly divided into 7 groups: (I) Control (S), (II) 5-ALA i.v. (S), (III) 5-ALA i.t. (S), (IV) CCPCA i.v. (S), (V) CCPCA i.t. (S), (VI) MN-CCPCA (S), and (VII) MN-CCPCA (M). The 4T1 tumor growth inhibition rates of 5-ALA i.t. (S), MN-5-ALA (S), CCPCA i.t. (S), MN-CCPCA (S), and MN-CCPCA (M) groups were calculated as 47.4%, 51.1%, 59.8%, 78.7%, and 92.3%, respectively (Fig. 7c, d). The low 5-ALA bioavailability and PpIX synthesis efficiency in the 5-ALA i.v. (S) group as evidenced by the metabolic FL images of PpIX (Fig. 7a, b) led to a poor therapeutic effect (Fig. 7c–f). Similarly, tumor recurrence occurred in all 4T1 tumor-bearing mice of 5-ALA i.t. (S), MN-5-ALA (S), CCPCA i.t. (S), and MN-CCPCA (S) groups (Fig. 7c–f). Nevertheless, the MN-CCPCA (M) group showed dramatic tumor growth inhibition within 2 weeks, indicating that intermittent irradiation in MN-CCPCA group can reduce the recurrence rate of 4T1 tumors. In addition, negligible change in the relative body weight of 4T1 tumor-bearing mice (Supplementary Fig. 11a) also proved that the optimized treatment strategy based on the MN-CCPCA patch has good biosafety.

Importantly, the in vivo immunofluorescence images of anti-GSH showed that all the treatment groups containing CCPCA could decrease the GSH level. The positive area ratio and positive area density of anti-GSH in 4T1 tumor treated with MN-CCPCA (M) were calculated as 0.31% and 0.0003, respectively (Supplementary Table 1). Besides, the HIF-1α expression levels in the CAT-loaded groups were lower than the other groups as the consumption of $O_2$ by 5-ALA-PDT without CAT caused more severe hypoxia (which may result in resistance to PDT in 4T1 tumors). It is worth noting that a higher expression level of HIF-1α was observed in the MN-CCPCA (S) group than the MN-CCPCA (M) group, indicating that intermittent irradiation is helpful to reduce tumor hypoxia (Fig. 7h and Supplementary Fig. 11b). By comparing H&E, TUNEL and Ki67 staining images, the nuclei in the MN-CCPCA (M) group exhibited the most severe karyopyknosis, the strongest apoptotic signal and negligible proliferative signal (Fig. 7h and Supplementary Fig. 11c, d). These results confirmed that the MN-CCPCA patch has efficient antitumor effects in a 4T1 tumor model when combining a microneedle transdermal delivery strategy with adjustable irradiation parameters under the FL/PA duplex imaging guidance.

## In vivo personalized theranostics for the U87MG tumor model

The US Food and Drug Administration (FDA) approval of 5-ALA was based on its utility as an intraoperative diagnostic tool rather than as a therapeutic for high-grade gliomas (HGGs)[2,8]. Effective clinical application of 5-ALA-PDT for HGGs needs to see improved accuracy and lower tumor recurrence rates. To this aim, we studied the metabolic dynamics of PpIX in U87MG tumor-bearing mice with different treatments. We saw that PpIX was synthesized throughout the body after i.v. administration of 5-ALA or CCPCA, especially in the brain (Fig. 8a, b), which may cause organelle-selective protein aggregation and phototoxicity to normal tissues as a higher concentration of PpIX than physiological condition[7,26,52]. Obvious FL signals of PpIX in both brain and normal skin tissues around tumors were also observed after i.t. administration of 5-ALA or CCPCA (Fig. 8a, b). The METP of MN-5-ALA group was recorded at 6 h post-administration, then the FL signals of PpIX rapidly cleared from the tumors, indicating short retention time of synthetic PpIX in tumor tissues. The partial amount of PpIX was nonspecifically distributed throughout the whole body due to the uncontrolled release of 5-ALA (Fig. 8a and Supplementary Fig. 12a–d). By contrast, PpIX was almost synthesized at the tumor sites and displayed a longer tumor retention time in the MN-CCPCA group (Fig. 8a, b and Supplementary Fig. 12a–d), as 6.87- and 5.73-fold higher FL intensities of PpIX were recorded at 24 h than those of the CCPCA i.t. and MN-5-ALA groups, respectively (Fig. 8a, b). In addition, the infiltration depth of PpIX in the MN-CCPCA group was more than 1.2 mm along the pierced direction of the tips with 1.96-fold and 1.24-fold increase of mean PpIX FL intensities across the skin than MN-5-ALA and CCPCA i.t. groups at 12 h, respectively (Fig. 8c and Supplementary Fig. 12e). These results supported that MN-CCPCA can deliver 5-ALA in a controlled manner with improved bioavailability than the other delivery modes (i.v., i.t.) of 5-ALA and CCPCA, as we also observed in a 4T1 breast tumor model (Fig. 7a, b and Supplementary Fig. 10a–d). Nevertheless, the METPs of PpIX in the A375, 4T1, and U87MG tumor models were recorded at 12, 24, and 12 h, respectively (Fig. 5a, Fig. 7a, and Fig. 8a), indicating that the heterogeneous metabolic dynamics of PpIX in different types of tumors are attributed to their intrinsic heterogeneity.

We next irradiated U87MG tumor-bearing mice with a 635 nm laser at 6, 12, and 24 h after the transdermal application of the MN-CCPCA patch according to the metabolic FL images of PpIX. The bioluminescence images of U87MG tumor-bearing mice before (day 0) and after treatment (day 14) suggested that all these administration modes of 5-ALA showed a certain tumor growth inhibition after laser irradiation (Fig. 8d and Supplementary Fig. 13a, b). The U87MG tumor growth inhibition rates of 5-ALA i.t. (S), MN-5-ALA (S), CCPCA i.t. (S), MN-CCPCA (S), and

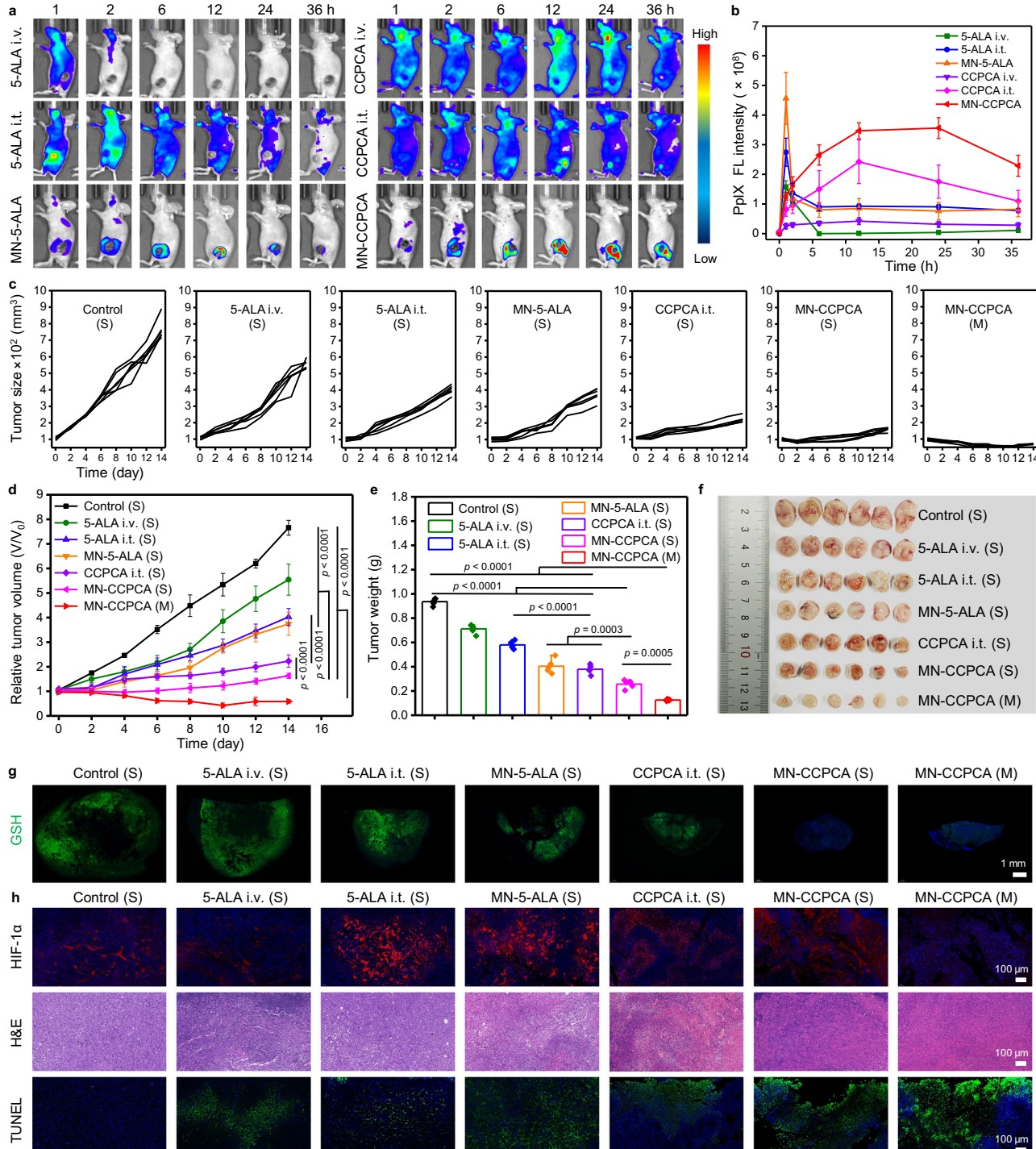

**Fig. 7 | Antitumor effects of MN-CCPCA patch in a 4T1 tumor model. a** In vivo PpIX FL images of 4T1 tumor-bearing mice after administration of 5-ALA i.v., 5-ALA i.t., MN-5-ALA, CCPCA i.v., CCPCA i.t., and MN-CCPCA patch at the indicated time points. **b** PpIX FL intensities of tumor tissues after different treatments at the indicated time points. Data represent the means ± SD from three biologically independent female 4T1 tumor-bearing mice, $n = 3$. **c** Individual tumor growth curves, **d** relative tumor volume growth curves, and **e** 4T1 tumor weight in different treatment groups on day 14, the symbols "S" indicate single laser irradiation and "M" indicate multiple laser irradiation. Data represent the means ± SD from six biologically independent female 4T1 tumor-bearing mice, $n = 6$. Statistical significance was calculated with two-tailed Student's $t$ test. **f** Photographs of excised 4T1 tumors ($n = 6$) on day 14 after different treatments. **g** In vivo immunofluorescence images of anti-GSH (green) and Hoechst (blue) in 4T1 tumors after different treatments. Scale bar = 1 mm. Images are from six biologically independent tumors of 6-week-old female BALB/c-nude mice, $n = 6$. **h** HIF-1α-, H&E-, TUNEL-stained sections of 4T1 tumors after various treatments indicated. Scale bar = 100 μm. Images are from six biologically independent 4T1 tumors of 6-week-old female BALB/c-nude mice, $n = 6$. Source data are provided as a Source data file.

MN-CCPCA (M) groups were calculated as 60.8%, 79.9%, 82.4%, 88.3%, and 95.2%, respectively (Fig. 8f, g). Same case as in 4T1 model, tumor recurrence occurred in both i.v. and i.t. paths of 5-ALA or CCPCA groups for U87MG tumors (Fig. 8d–f and Supplementary Fig. 13a, b). However, the MN-CCPCA (S) and the MN-CCPCA (M) groups did not show tumor recurrence within 2 weeks and even displayed complete elimination of U87MG tumor (Fig. 8d–g and Supplementary Fig. 13a, b).

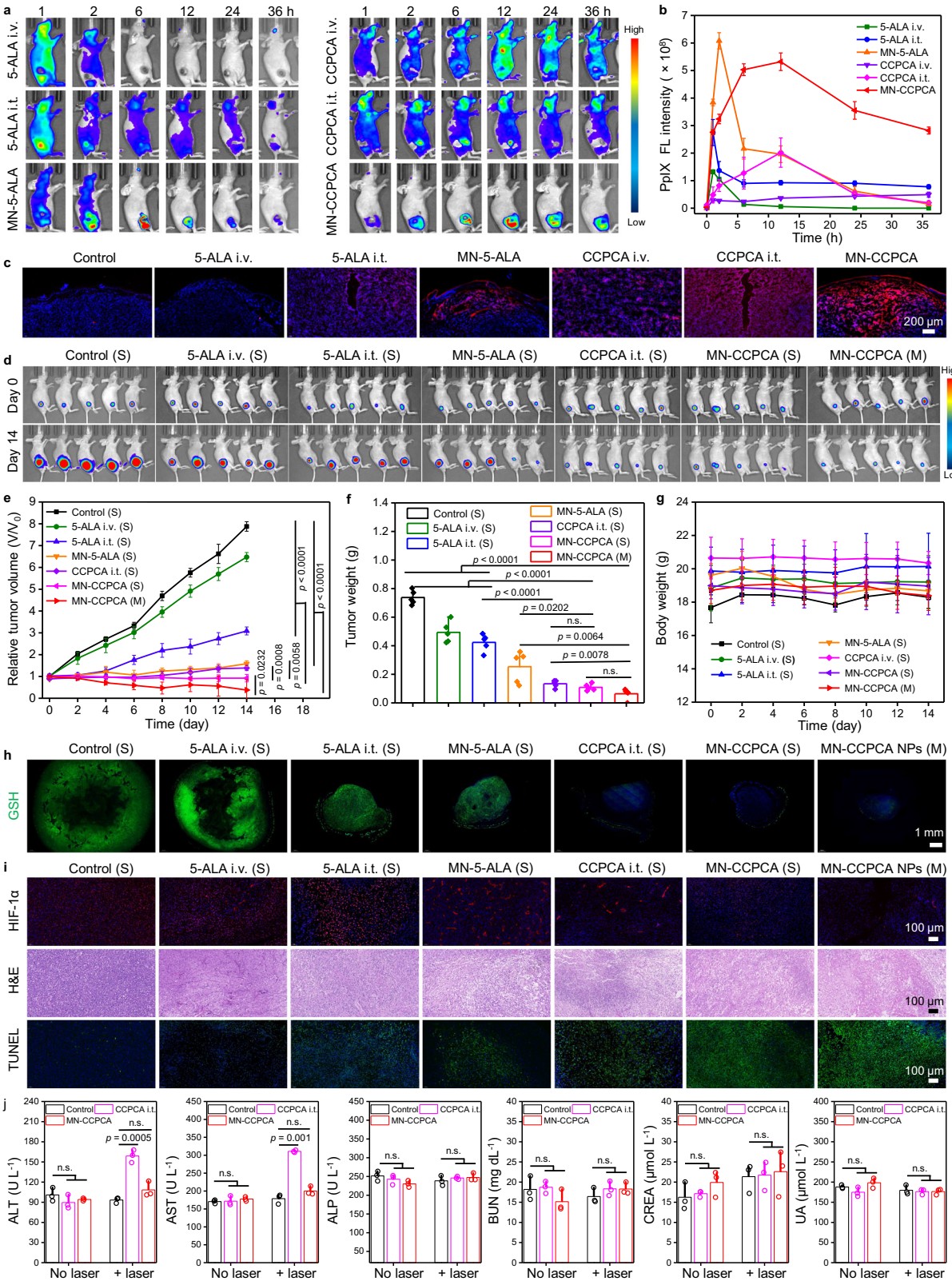

Consistent with our previous results of 4T1 tumor sections, we found that groups treated with CCPCA could deplete GSH, resulting in small tumor area, low positive area ratio, and positive area density of anti-GSH FL signals in U87MG tumors (Fig. 8h and Supplementary Table 1). The hypoxic microenvironment in U87MG tumors was significantly reversed in the treatment groups contained CCPCA according to HIF-1α staining images (Fig. 8i and Supplementary

Fig. 13c). Compared to the lower apoptosis FL signal of fluorescein-dUTP in other groups, the higher fully covered FL signal of fluorescein-dUTP in the MN-CCPCA (M) group revealed the most intense signal of apoptosis in tumors (Fig. 8i and Supplementary Fig. 13d). Severely pyknotic nuclei and negligible Ki67 expression were observed in MN-CCPCA (M) groups (Fig. 8i and Supplementary Fig. 13e, f).

**Fig. 8 | Enhanced therapeutic efficacy and biosafety of MN-CCPCA patch for U87MG tumors. a** In vivo PpIX FL images of U87MG tumor-bearing mice after administrations of 5-ALA i.v., 5-ALA i.t., MN-5-ALA, CCPCA i.v., CCPCA i.t., and MN-CCPCA patch at the indicated time points. **b** PpIX FL intensities of tumor tissues after different treatments at the indicated time points. Data represent the means ± SD from three biologically independent female U87MG tumor-bearing mice, $n = 3$. **c** Colocalization FL images of intratumoral PpIX by cross sections after different treatments at 12 h post-treatment (red FL signal from PpIX and blue FL signal from Hoechst), scale bar = 200 μm. Data represent the means ± SD from three biologically independent female U87MG tumor-bearing mice, $n = 3$. **d** Bioluminescence images of U87MG tumor-bearing mice ($n = 5$) treated with 5-ALA i.v., 5-ALA i.t., MN-5-ALA, CCPCA i.t., MN-CCPCA (S), MN-CCPCA (M) on days 0 and 14. The symbols "S" indicate single laser irradiation and "M" indicate multiple laser irradiation. **e** Relative tumor volume growth curves, **f** tumor weight, and **g** body

weight in different treatment groups on day 14. Data represent the means ± SD from five biologically independent female U87MG tumor-bearing mice, $n = 5$. Statistical significance was calculated with two-tailed Student's $t$ test. **h** In vivo immuno-fluorescence images of anti-GSH (green) and Hoechst (blue) in 4T1 tumors after different treatments, scale bar = 1 mm. Images are from five biologically independent tumors of 6-week-old female BALB/c-nude mice, $n = 5$. **i** HIF-1α-, H&E-, and TUNEL-stained sections of U87MG tumors after various treatments indicated, scale bar = 100 μm. Images are from five biologically independent tumors of 6-week-old female BALB/c-nude mice, $n = 5$. **j** Serum biochemical analysis of U87MG tumor-bearing mice in CCPCA i.t. and MN-CCPCA groups before and after 30 min laser irradiation. Data represent the means ± SD from three biologically independent blood samples of 6-week-old female U87MG tumor-bearing mice, $n = 3$. Statistical significance was calculated with two-tailed Student's $t$ test. Source data are provided as a Source data file.

As we know PpIX can be activated by UV or visible light, causing systemic porphyria and damage to normal tissues[7], the uncontrolled release of 5-ALA and nonspecific distribution of PpIX by i.v. or i.t. injections of CCPCA raised biosafety concerns for following irradiation treatment. Although there was no evident damage to the major organs of U87MG tumor-bearing mice after i.v., i.t. or MN administrations of CCPCA within 14 days (Supplementary Fig. 14), we further conducted serum biochemical analysis to assess the biosafety of i.t. or MN administrations of CCPCA before and after laser irradiation. As shown in Fig. 8j, there was no significant difference in serum biochemical parameters without laser exposure. However, alanine aminotransferase (ALT) and aspartate aminotransferase (AST) significantly increased in CCPCA i.t. group after 30 min of laser irradiation. In contrast, there was negligible difference in serum biochemical parameters of liver (including ALT, AST, and alkaline phosphatase (ALP)) and kidney function (including urea nitrogen (BUN), creatinine (CREA), and uric acid (UA)) of mice treated with MN-CCPCA patch before and after 30 min of laser irradiation, suggesting that MN administration of CCPCA can improve the biosafety of 5-ALA-PDT.

Taken together, co-delivery of 5-ALA and CAT via MN-CCPCA patch offers a strategy to upregulate intratumoral PpIX biosynthesis, monitor in vivo real-time PpIX and sO$_2$ levels, and optimize treatment parameters through duplex FL and PA imaging. This approach overcomes heterogeneous PpIX metabolism in different type of solid tumors and ultimately improves therapeutic efficacy and safety.

## Discussion

The method of providing 5-ALA exogenously to enhance PpIX accumulation is restricted by the inefficient delivery strategies of 5-ALA and poor biological accumulation efficiency of PpIX[3,7,8]. Such physiological PpIX metabolic kinetics limits 5-ALA-PDT of malignant tumors in a hypoxic microenvironment, as conventional 5-ALA-PDT for malignant tumors usually induces more serious recurrence and metastasis[4,5,53]. Here, we show that reversing the hypoxic tumor microenvironment by continuously catalyzing endogenous H$_2$O$_2$ into O$_2$ to elevate the intratumoral sO$_2$ level can increase the in vivo biosynthesis of PpIX through the regulation of PpIX-related synthetases. Inspired by this concept, we developed a 5-ALA and CAT co-delivery system based on a theranostic microneedle patch (MN-CCPCA) that forces cancer cell to release the false signal of a "heme deficiency" and thus shuts down the outflow of PpIX through the cascaded downregulation of HIF-1α and FECH. The overall result is a halt to the negative feedback on exogenous 5-ALA-induced PpIX biosynthesis and instead, the upregulation of endogenous PpIX biosynthesis. Our in vitro and in vivo analyses supported that the tumor retention time of PpIX was significantly prolonged in A375, 4T1, and U87MG tumors after application of the MN-CCPCA patch, and the FL intensity of PpIX increased throughout the period compared with i.v. i.t. or MN administration of 5-ALA. We thus had an ample theranostic time window for 5-ALA-PDT with a high efficiency as we could monitor the real-time metabolic kinetics of PpIX

and O$_2$. This will also greatly enrich the clinician's selectivity for treatment parameters and open the treatment gate for more potential indications of 5-ALA-PDT.

The conversion of 5-ALA to PpIX depends on complex regulatory factors, such as temperature, pH, the activity of rate-limiting enzymes, and the number of mitochondria. Methods to block the conversion of PpIX to heme by iron chelators or FECH inhibitors have become more efficient and are fully validated to enhance the concentration of PpIX[7,24,27]. However, downregulation of FCCH by HIF-1α inhibitors is a slow process while the metabolism of PpIX in vivo is very fast. As such, both the continuous supply of O$_2$ from the catalytic homolysis of endogenous H$_2$O$_2$ and the sustained release of 5-ALA are essential. Microneedles can directly penetrate the stratum corneum for drug delivery[54–57], while nanomedicines can facilitate sustain drug release and ensure multifunctional delivery[48,58,59]. We thus hypothesized that a transdermal MN patch combining 5-ALA and CAT co-loaded CCPCA NPs could potentially upregulate the biosynthesis of PpIX and improve its tumor retention in vivo. These reasons served as our rationale to develop this microneedle-based CAT and 5-ALA co-loaded patch to extend the METP of intratumoral PpIX. However, it should be noted that the influence of intracellular molecular oxygen on PpIX metabolism is multi-faceted, and more abundant and precise means are needed to characterize the biochemical changes caused by the generation of molecular oxygen inside cancer cells, such as to demonstrate whether it is involved in epigenetic modification of heme synthesis through multi-omics analysis (proteomics, genomics, transcriptomics, metabolomics) or to confirm whether molecular oxygen directly affects the supply of Fe$^{2+}$ in mitochondria through chemical reactions.

Although PDT has been clinically applied for dozens of cancer indications, the inherent drawbacks of PDT restrict its further development. PDT combined with other therapies can exploit the advantages and offset the disadvantages of each therapeutic modality[4,58]. In our work, the induction of severe oxidative stress by 5-ALA-PDT and bioactive ion interferences (Ca$^{2+}$/Cu$^{2+}$) were complementary. On the one hand, the MN-CCPCA patch could simultaneously provide sufficient O$_2$ and upregulate the accumulation of PpIX, which greatly increased the generation of $^1$O$_2$. On the other hand, the stronger oxidative stress caused by 5-ALA-PDT promotes therapeutic effects through the cascade reactions of IIT. Both in vitro and in vivo experiments verified the synergistic effect of IIT and 5-ALA-PDT.

Generally, the therapeutic efficiency of 5-ALA-PDT is related to the concentration of O$_2$ and PpIX, the intensity and duration of irradiation, and even the sensitivity of tumor tissues of different pathological grades[3,21,22]. Optimization of these parameters is difficult due to the lack of a precise and personalized theranostic platform. While the diversity in 5-ALA administration methods and preparation formulations will lead to different tumor accumulation peaks of PpIX, the biodistribution of PpIX, metabolic kinetics, and intratumoral sO$_2$ levels before and after laser exposure are also variable due to tumor heterogeneity. We therefore propose an ideal theranostic platform for the

determination of irradiation parameters based on intratumoral sO$_2$ and PpIX metabolic kinetics in real time and look forward to providing a more effective diagnosis and treatment method for the clinical application of 5-ALA-PDT.

## Methods

### Materials

Calcium chloride (CaCl$_2$) and copper chloride (CuCl$_2$) were obtained from Shanghai Macklin Biochemical Co., Ltd. CAT (40,000–60,000 units/mg protein, 250 kDa), bovine serum albumin, 5-aminolevulinic acid hydrochloride, and 2',7'-dichlorodihydrofluorescein diacetate were obtained from Sigma-Aldrich. 1-(3-Dimethylaminopropyl)-3-ethylcarbodiimide hydrochloride (EDC·HCl) was purchased from Energy Chemical (Shanghai). N-hydroxysuccinimide (NHS) and RhB were obtained from J&K Scientific (Beijing). Hyaluronic acid sodium salt (100 kDa) and super active hyaluronic acid (5 kDa) were purchased from Hefei BOSF Biotechnology Co., Ltd. (China). The customized PDMS molds were provided by Guangzhou LAKEY Micromachining Center (China). Dulbecco's modified Eagle's medium (DMEM) with/without sugar and 0.25% trypsin-EDTA were obtained from Gibco® (USA). A Free Protoporphyrin Detection Kit was purchased from Beijing Leagene Biotechnology Co., Ltd. Cell Counting Kit-8 (CCK-8), GSH and GSSG Assay Kit, Mitochondrial Membrane Potential Assay Kit, Hydrogen Peroxide Assay Kit, and Catalase Assay Kit were obtained from Beyotime Biotechnology (China). Fetal bovine serum, Hoechst 33342, and a Fluo-4 AM Calcium Assay Kit and anti-ALAS1 (catalog no. PA5-100995) were obtained from Thermo Fisher Scientific (Waltham, USA). Anti-α Tubulin (catalog no. ab52866), anti-HIF-1α Alexa Fluor ® 647 (catalog no. ab203848), and anti-GSH (catalog no. ab19534) were purchased from Abcam. Anti-Ki67 (catalog no. GB111499) and TUNEL (G1510) were purchased from Servicebio. Anti-HIF-1α (catalog no. 14179S and 36169T) and GAPDH (catalog no. 2118S) were purchased from Cell Signaling Technology. FECH (catalog no. NBP2-33413) and anti-caspase-3 p11 Alexa Fluor ® 488 (catalog no. sc-271759) were purchased from Novus Biologicals and Santa Cruz Biotechnology, respectively. All cell lines (4T1, 293T, A375, luciferase U87MG) were purchased from Cell Bank of Chinese Academy of Sciences (Shanghai, China).

### Characterizations

NPs morphologies were visualized by TEM (HT7700, Hitachi Electronics). The size distributions and zeta potentials of different NPs were detected with a Malvern Zetasizer Ultra. Elemental mapping analysis was performed on a field emission TEM (JEM-F200, Japan). UV-vis spectra were recorded by a UV-vis spectrophotometer (Agilent Cary 60, USA). Thermogravimetric analysis (TGA) was performed on a thermal analyzer (Mettler Toledo, Switzerland). The proportion of ions (Ca$^{2+}$, Cu$^{2+}$) in the NPs was detected by an ICP-AES (JY 2000-2, France). The morphologies of the MN patches were observed by stereo microscopy (Nikon SMZ18, Japan) and SEM (Thermo Scientific, FEI-Apreo).

### Preparation of NPs

CCP NPs were prepared as previously described but with minor modifications[12,42]. In brief, 100 mg of BSA was dissolved in 10 mL glucose-free DMEM, and then 100 μL CuCl$_2$ was added (0.1 M). The system was sealed and stored at 37 °C for 24 h. Subsequently, 100 μL CaCl$_2$ (1 M) was added to the system and further stored at 37 °C for 24 h. Finally, the resulting NPs were separated and collected by repeated centrifugation (15,285 × $g$, 10 min) with deionized (DI) water and lyophilized for subsequent experiments.

For CCPC NPs synthesis, 2 mg CCP NPs was dispersed into 2 mL 0.1 mM PBS buffer (pH adjusted to 7.4), and then 2 mg EDC·HCl was directly dissolved in 100 μL DMSO and added to the above mixture.

After stirring in the dark for 1 h at room temperature, 2 mg NHS was dissolved in 100 μL DMSO and added to the reaction mixture. Then, 0.2 mg CAT was added and stirred for 6 h at 37 °C to obtain CCPC NPs. The obtained products were collected by centrifugation at 15,285 × $g$ for 10 min. The RhB-loaded CCPC NPs (CCPC-RhB NPs) were fabricated by adding RhB, CAT, and CCP NPs at a feeding ratio of 1:1:10.

To synthesize CCPCA NPs or CCPCA-RhB NPs, 2 mg CCPC NPs or CCPC-RhB NPs were dispersed in 2 mL 0.1 mg mL$^{-1}$ 5-ALA solution (0.1 mM PBS solution, pH adjusted to 7.0), followed by gentle stirring in the dark for 2 h at room temperature. Subsequently, the resulting products were separated and collected by centrifugation at 15,285 × $g$ for 10 min. The absorbance of the supernatant at 200 nm was detected by UV-vis spectrophotometry to calculate the unloaded 5-ALA. The drug encapsulation efficiency and loading efficiency were calculated according to the following equations:

$$\text{Encapsulation efficiency} = (m_{\text{total 5-ALA}} - m_{\text{unloaded 5-ALA}})/m_{\text{total 5-ALA}} \times 100\%$$

(1)

$$\text{Loading efficiency} = (m_{\text{total 5-ALA}} - m_{\text{unloaded 5-ALA}})/m_{\text{total NPs}} \times 100\%$$

(2)

### In vitro degradation

For the pH-dependent degradation study, 1 mL CCPCA NPs suspension (3 mg mL$^{-1}$) was added to a dialysis bag (MWCO = 3.5 kDa) and then dialyzed in an additional 9 mL PBS buffer solutions with varying pH values (7.4 and 6.0). The above mixture was placed in a shaker at 37 °C to simulate the different internal environments. Afterward, 1 mL medium was collected at different time points to determine the amount of Cu$^{2+}$ and Ca$^{2+}$ ions by ICP analysis.

### CAT activity assay

The catalytic activity of CCPCA NPs was quantitatively analyzed by UV-vis spectrophotometry. 10 mM H$_2$O$_2$ was treated with DI water, 0.1 μg mL$^{-1}$ CAT and 1.2 μg mL$^{-1}$ CCPCA NPs. The absorption profile of H$_2$O$_2$ at 240–330 nm was then recorded by UV-vis spectrophotometry at the indicated time points.

The O$_2$ generation capacity of CAT was determined by adding different concentrations of CCPCA NPs (0, 20, 40, 80, 160 μg mL$^{-1}$) into 100 μM or 1 mM H$_2$O$_2$ solution. Then, the dissolved O$_2$ in different solutions was measured using a portable dissolved oxygen meter (Rex, JPBJ-608, China).

### Preparation of MN patches

A two-step micromolding process was used to prepare the NPs-loaded MN patches[54]. Briefly, the NPs were dispersed in 600 μL distilled water. Then, 100 μL NPs suspension (10 mg mL$^{-1}$) was deposited on the surface of each PDMS mold through a pipette. The device was placed in a vacuum drying oven and continuously vacuumized five times (3 min each). Then, HA solution (180 mg mL$^{-1}$, sodium hyaluronate: super active hyaluronic acid = 1:5, w/w) was added to the surface of each PDMS mold. The device was dried at 37 °C for 13 h and then stored in a dryer in the dark.

### Mechanical strength test

The mechanical strength of the MN patch was tested on a tensile load frame. The tension was continuously monitored on a stress–strain gauge using a steel plate pressing the MN patch along the y-direction. The top of the steel plate moved toward the MN patch at a rate of 1 mm s$^{-1}$. When the needle began to bend, the destructive force of the MN patch was recorded.

## Skin penetration test

To prove the penetration ability of the MN patches, the prepared MN patches were topically applied to the skin of mice and rats for 10 min. After removing the patch, the exposed skin area was stained with MB for 10 min. Before stereo-microscopic observation, the remaining dye was wiped off the skin surface with tissue paper. For histological staining, the mouse skin treated with different MN patches was fixed with 10% tissue fixation solution. For 3D reconstructed imaging of RhB FL on tumor skin, the CCPCA-RhB NPs were loaded in the patch and then applied to the A375 tumor-bearing mice for 10 min. The FL image was observed after 4 h under a multiphoton microscope (NIKON-A1MP) at 540 nm excitation and 625 nm emission.

## Cellular experiments

293T, A375, 4T1, and luciferase U87MG cells were cultured in DMEM containing 10% fetal bovine serum and 1% penicillin–streptomycin and placed in an incubator at 37 °C with 5% $CO_2$. All cells were cultured in 96-well plates at a density of $1 \times 10^4$ cells/well for dark cytotoxicity and phototoxicity assays. A CCK-8 assay kit was used to detect cell viability.

All FL images were captured using an Operetta CLS™ High Content Analysis System (HCAS) (Operetta, PerkinElmer, USA). A375 cells were seeded in a 96-well Ultra CellCarrier™ microplate (PerkinElmer) equipped with a temperature (37 °C) and $CO_2$ (5%) control option (TCO).

To test the $O_2$ production capacity of CCPCA NPs, A375 cells were seeded at a density of $8 \times 10^3$ cells/well and incubated with RDPP (0.1 μM) and Hoechst (1 μg $mL^{-1}$) for 20 min. Subsequently, 100 μL of a blank medium, CCP, CCPA, CCPC, and CCPCA (80 μg $mL^{-1}$) were added and further incubated for 24 h in the dark. Thereafter, the HCAS was used to capture FL images under excitation at 485 nm and emission at 545 nm.

For cellular PpIX biosynthesis, A375 cells were seeded at a density of $1 \times 10^4$ cells/well and then exposed to equivalent amounts of free 5-ALA or NPs (CCP, CCPC, CCPA, CCPCA) for 24 h in the dark. Then, the in vitro FL signal of PpIX was recorded at 405 nm excitation and 635 nm emission with a 1-h interval time.

For $[Ca]_{CYT}$ detection, A375 cells were seeded at a density of $8 \times 10^3$/well for 24 h and incubated with blank 5-ALA, CCP, CCPA, and CCPCA (5.4 μg $mL^{-1}$ 5-ALA, 80 μg $mL^{-1}$ NPs) for another 24 h in the dark. The cells were washed twice with PBS and cultured with Fluo-4 AM (1 μM) and Hoechst (1 μg $mL^{-1}$) for 20 min at 37 °C. Subsequently, the cells were either irradiated with a 635 nm laser (200 mW $cm^{-2}$, 5 min) or cultured in the dark. Finally, the green FL of $[Ca]_{CYT}$ was acquired by HCAS under excitation at 488 nm and emission at 520 nm.

For ROS detection, A375 cells were seeded at a density of $1 \times 10^4$ cells/well and incubated with 100 μL DMEM containing 5-ALA, CCPA, and CCPCA NPs (at an equivalent concentration of 5.4 μg $mL^{-1}$ 5-ALA). After culturing for another 24 h, the cells were washed three times with PBS and then stained in 100 μL serum-free DMEM supplemented with DCFH-DA (10 μM) and Hoechst (1 μg $mL^{-1}$) at 37 °C for 1 h. Subsequently, the cells were washed twice with PBS, irradiated with a 635 nm laser (200 mW $cm^{-2}$, 5 min), or cultured in the dark. The cells were immediately analyzed with HCAS at 470 nm excitation and 535 nm emission.

For $\Delta\Psi$m evaluation, A375 cells were seeded at a density of $1 \times 10^4$ cells/well for 24 h and then incubated with 5-ALA, CCPA, and CCPCA NPs (at equivalent 5.4 μg $mL^{-1}$ 5-ALA) for another 24 h. Subsequently, the cells were irradiated with a 635 nm laser (200 mW $cm^{-2}$, 5 min). After incubation for another 30 min, the cells were stained with Hoechst (1 μg $mL^{-1}$) and JC-1 (1 μg $mL^{-1}$). Green FL images of the monomer were acquired at a 490 nm excitation wavelength and 530 nm emission wavelength. The FL images of J-aggregate were acquired at 525 nm excitation wavelength and 590 nm emission wavelength.

To detect the levels of intracellular PpIX, GSH, and $H_2O_2$, A375 cells were seeded in a 6-well plate at a density of $2 \times 10^5$ cells/well for 24 h and incubated with blank medium, CCP, CCPA, CCPC, and CCPCA NPs for an additional 24 h. Then, the cells were washed three times with PBS and collected. The levels of intracellular PpIX, GSH, and $H_2O_2$ were detected using a Free Protoporphyrin Detection Kit, GSH Assay Kit, and Hydrogen Peroxide Assay Kit according to the supplier's instructions.

## Immunofluorescence in vitro

To detect the change of HIF-1α expression, A375 cells were seeded at a density of $5 \times 10^3$ cells/well for 24 h under hypoxic (1% $O_2$, 5% $CO_2$, and 94% $N_2$) and normoxic (21% $O_2$, 5% $CO_2$, and 74% $N_2$) conditions, respectively, and then incubated with blank medium, 5-ALA, CCPA, and CCPCA NPs for another 24 h. Antibodies against mouse monoclonal HIF-1α Alexa Fluor ® 647 (1:1000, Abcam, catalog no. ab203848), rabbit monoclonal to anti-alpha Tubulin (1:1000, Abcam, catalog no. ab52866) were chosen for immunofluorescence without laser irradiation according to the supplier's instructions. Mouse monoclonal HIF-1α Alexa Fluor ® 647 (1:1000, Abcam, catalog no. ab203848) and caspase-3 p11 Alexa Fluor ® 488 (1:500, Santa Cruz Biotechnology, catalog no. sc-271759) were chosen for immunofluorescence after laser irradiation (200 mW $cm^{-2}$, 5 min).

## Western blot

A375 cells were lysed via vigorous sonication in an ice bath. Then, the protein concentrations were determined, followed by immunoblotting of cell lysates using antibodies against HIF-1α (1:1000, Cell Signaling Technology, catalog no. 14179S and 36169T), FECH (1:1000, Novus Biologicals, catalog no. NBP2-33413), ALAS1 (1:1000, Thermo Fisher Scientific, catalog no. PA5-100995), and GAPDH (1:1000, Cell Signaling Technology, catalog no. 2118S) according to standard protocols. A chemiluminescence imaging system was used to monitor the expression levels of different proteins (Fluor Chem E).

## In vivo FL and PA duplex imaging

To analyze the real-time metabolism of PpIX, in vivo FL imaging of PpIX was performed using an IVIS Spectrum system (PerkinElmer). When the tumor volume reached ~100 $mm^3$, the A375 tumor-bearing mice were randomly divided into five groups ($n = 3$) to detect the metabolic dynamics of PpIX after MN administration: I. Control (blank MN), II. MN-CCP, III. MN-CCPC, IV. MN-CCPA, V. MN-CCPCA. Similarly, the U87MG/4T1 tumor-bearing mice were randomly divided into 7 groups ($n = 3$) to detect the metabolic dynamics of PpIX by different 5-ALA and CCPCA delivery methods: 1. Control, 2. 5-ALA i.v., 3. CCPCA i.v., 4. 5-ALA i.t., 5. CCPCA i.t., 6. MN-5-ALA, 7. MN-CCPCA. All mice were imaged at different time intervals (0, 2, 6, 12, 24, and 36 h). To observe the biodistribution of intratumoral PpIX, the tumors from every group were collected at the METP of PpIX and rapidly frozen for FL cross-sections.

For PA imaging, when the tumor volume reached ~100 $mm^3$, the tumor area was imaged under a Vevo LAZR-X imaging system (Fujifilm Viualsonic, Toronto, Canada) at different intervals after application of MN patches: 1 h, 24 h (before irradiation), and 48 h (after 24 h post-irradiation). All the groups ($n = 3$) were irradiated with a 635 nm laser (200 mW $cm^{-2}$, 10 min) after determining the intratumoral $sO_2$ at 24 h.

## Tumor mouse model and immunofluorescence staining

All animal experiments were conducted in compliance with the Institutional Animal Care and Use Committee of Shenzhen University (AEWC-SZU, maximal tumor size: <2000 $mm^3$). Four-week-old female BALB/c-nude mice and rats were obtained from Guangdong Medicinal Laboratory Animal Center (China). All of the experimental mice and rats were housed under standard conditions (temperature: ~22 °C,

humidity: 40–70%, 12 h dark-light cycles) with free access to sterile food and water. To confirm MN-CCPCA patch-mediated synergistic IIT and 5-ALA-PDT, A375 cells ($5 \times 10^6$ cells/mouse) were subcutaneously inoculated into the right dorsal flank of all mice. When the tumor size reached ~70 mm³, the mice were randomly assigned to six groups ($n = 5$): I. Blank MN, II. MN-CCP, III. MN-CCPC, IV. MN-CCPCA, V. MN-CCPA ( + ), and VI. MN-CCPCA ( + ). All groups were administered an equivalent dose of NPs (50 mg kg⁻¹) and 5-ALA (3.4 mg kg⁻¹). After all mice were anesthetized, the MN patches were topically applied to the tumor area for 10 min. For the laser treatment group, the mice were irradiated with a 635 nm laser (200 mW cm⁻², 10 min each time) at 6, 12, and 24 h. The tumor size and body weight were measured every 2 days. The tumor volumes were calculated according to the formula: volume = width² × length/2. The tumors in different groups were harvested for Ki67 (catalog no. GB111499, Servicebio), TUNEL (G1510, Servicebio), and HIF-1α (catalog no. 36169T, Cell Signaling Technology) immunofluorescence staining after laser irradiation at 24 h post-treatment. Major organs of A375 tumor-bearing mice were harvested for H&E staining on 14th day.

To determine the MN-CCPCA patch could effectively treat solid malignant tumors, two different malignant tumors, U87MG and 4T1, were selected. U87MG and 4T1 cells ($1 \times 10^6$ cells/mouse) were subcutaneously inoculated into the dorsal flank of mice. When the tumor size reached ~100 mm³, the mice were randomly assigned to seven different groups as follows: I. Control (S), II. 5-ALA i.v. (S), III. 5-ALA i.t. (S), IV. MN-5-ALA (S), V. CCPCA i.t. (S), VI. MN-CCPCA (S), VII. MN-CCPCA (M). All groups were administered an equivalent dose of 5-ALA (3.4 mg kg⁻¹). The U87MG tumor-bearing mice of MN-CCPCA (M) group were irradiated at 6, 12, and 24 h, the 4T1 tumor-bearing mice of MN-CCPCA (M) group were irradiated at 12, 24, and 36 h (200 mW cm⁻², 10 min each time). The other treatment groups received a single irradiation (200 mW cm⁻², 30 min) at the METP of PpIX according to its metabolic kinetic analysis. Blood was collected from mice of Control, CCPCA i.t, and MN-CCPCA groups before and after laser irradiation (200 mW cm⁻², 30 min) for analysis. U87MG and 4T1 tumors were harvested for GSH (catalog no. ab19534, Abcam) immunofluorescence staining on day 14. The same process of other immunofluorescence staining was followed as the treatment process of A375 tumor-bearing mice. The positive area ratio and positive area density of GSH FL signals were calculated according to the following equations:

$$\text{Positive area ratio} = s_{\text{positive area}} / s_{\text{tissue area}} \times 100\% \quad (3)$$

$$\text{Positive area density} = I_{\text{integrated optical density}} / s_{\text{tissue area}} \times 100\% \quad (4)$$

### Statistical analysis
All data represent the means ± SD. Statistical differences were determined by an unpaired two-tailed Student's $t$ test between two groups using Excel 2020 software or one-way analysis of variance between multiple groups (ANOVA) by GraphPad Prism 8.0 (GraphPad Software, Inc., CA, USA) software when appropriate. A $p$ value <0.05 was considered to be statistically significant.

### Reporting summary
Further information on research design is available in the Nature Research Reporting Summary linked to this article.

### Data availability
The authors declare that the experimental data supporting the findings of this study are available within the Article, Supplementary Information, or Source data file. Source data are provided with this paper.

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

## Acknowledgements

This work was financially supported by the National Key R&D Program of China (2020YFA0908800, 2018YFA0704000), the National Natural Science Foundation of China (82071985), the Basic Research Program of Shenzhen (JCYJ20200109105620482, JCYJ20180507182413022), and the Shenzhen Science and Technology Program (KQTD20190929172538530). The authors would like to thank the Instrumental Analysis Center of Shenzhen University (Lihu Campus).

## Author contributions

P.H. conceived and designed the study. G.H. and Y.S.L. performed the experiments and analyzed the results. G.H. and M.R.Y. drew all figures and wrote the manuscript. L.H.F., H.T., and S.L. revised the manuscript. P.H. and L.J. finalized the manuscript, provided funding, and supervised this project.

## Competing interests

The authors declare no competing interests.
