## [Peer Review File · Nature Communications]

Synthetic biology-instructed transdermal microneedle patch for traceable photodynamic therapyReviewers' comments:

Reviewer #1 (Remarks to the Author): with expertise in 5-ALA and PDT therapy

In this manuscript, the authors developed a transdermal theranostic microneedle patch integrated with 5-ALA and catalase (CAT)-coloaded tumor acidity-responsive copper-doped calcium phosphate nanoparticles for efficient 5-ALA-PDT. The preparation and characterizations of MN-CCPCA patch was well arranged and studied. The synergistic effect of oxygen generation, GSH consumption and ion-interference therapy maximized the efficacy of 5-ALA-PDT, which was investigated in several solid tumors. However, the concept of oxygen production and ion-interference enhanced PDT has been previously reported in many literatures, thus this work lacks of sufficient novelty. In addition, in vivo fluorescence/photoacoustic duplex imaging guided therapy is not supported by solid evidences. The main strength of the work is that the enhanced therapeutic efficacy was systemically studied; however, synergistic antitumor mechanism was not investigated and the results could not fully support the authors' claims. Therefore I can't recommend publication of this work at NC. There are also other questions that need to be addressed if this work is going to be published elsewhere.

Major concerns:

1. In the method to measure O₂ generation, PpIX production, and expression of HIF-1 α and FECH in A375 cells placed in incubator at 37 °C with 5% CO₂, it is strange to observe such a difference in differently treated A375 cells. The normoxic environment would also provide enough oxygen level, and affect the detection results. Why did not the authors conduct the experiment in hypoxia environment?

2. In the part of in vivo PpIX metabolism, tumor-bearing mice were treated with MN-CCPCA for only 36 h, and an enhanced production of PpIX metabolism was observed. However, it is a little surprising to see that catalase could alleviate tumor hypoxia microenvironment in tumor-bearing mice and even affected the protein expression (HIF-1 α and FECH) in such a short time (36 h in antitumor study). Details investigations need to be provided.

3. In the part of oxygen production, catalase catalyzed tumor hydrogen peroxide to produce oxygen. As far as we know, catalase temporarily alleviates hypoxia tumor environment, which has been widely used for enhanced dynamic therapy. However, because the hydrogen peroxide is usually limited in tumor tissues, how did it sustainably produce oxygen? The concentration change of tumor hydrogen peroxide should be also evaluated after treatment with MN-CCPCA. The tumor oxygen level should be measured for at least one week to support the claim of "sustainable oxygen saturation".

4. In the antitumor treatment, why were the mice irradiated at 6, 12 and 24 h post-transdermal application of patches. Is there any evidence to support the irradiation for three times, as the tumor oxygen level could be reduced after each PDT treatment.

5. An enhanced antitumor therapeutic efficacy of MN-CCPCA was detected in different tumor models (breast cancer, glioblastoma, melanoma). The authors declared the O₂ generation by PpIX along with Cu²⁺/Ca²⁺ released by CCPCA NPs not only consumed antioxidant glutathione but also achieved stronger cytosolic free Ca²⁺ ([Ca]_{CYT}) overload for ion-interference enhanced repeatable 5-ALA-PDT. However, only antitumor result was shown in this study, and there was no mechanism study conducted to support this claim at molecular level.

6. In Fig. 7 and Fig. 8, 12, 24, and 36 h were selected as the irradiation times (200 mW cm⁻², 10 min each time) for the MN-CCPCA (+) group, while the other treatment groups were irradiated for only one time at the METP of PpIX according to its metabolic kinetic analysis. The comparison of therapy efficacy here would not be fair.

Minor concerns:

7. In Fig. 5d and 5e, MN-CCP and MN-CCPA should be studied.

8. In line 50-51, "a short treatment window and a short treatment window" should be revised, etc.

Reviewer #2 (Remarks to the Author): with expertise in microneedles and NPs

In this manuscript, Huang and coworkers leveraged a synthetic biology approach to construct a transdermal theranostic microneedle patch integrated with 5-Aminolevulinic acid (5-ALA) and catalase (CAT)-coloaded calcium phosphate nanoparticles for efficient 5-ALA-PDT. The author demonstrated that the continuous oxygen generation by CAT in vivo reverses tumor hypoxia and could promote protoporphyrin IX (PpIX) accumulation. Besides that, the in vivo fluorescence/photoacoustic duplex imaging could monitor intratumoral oxygen saturation and PpIX metabolic kinetics simultaneously. However, the advantage of delivery routing by microneedle compared with intratumor injection in this study is unclear. The concept of "Synthetic biology-instructed..." seems oversold here. Overall, this manuscript needs a major revision before potential acceptance.

1. The authors need to clarify the advantage and necessity of a microneedle patch for the nano-mixture delivery in this study, and another control group, intratumor injection of the therapeutic agents, should be added.

2. In this article, the authors demonstrate that Cu produces more reactive oxygen species and enhances tumor killing by consuming glutathione, but this conflicts with the role of CAT, which may lead to higher levels of reactive oxygen species and inhibit the production of PpIX. The authors need more evidence to explain the internal relationship between them.

3. The results of reactive oxygen species in the non-PDT treatment group should be proved because Cu also induces the production of highly reactive oxygen species.

4. Immunofluorescence in Figure 7 shows that Ki67 is not located in the nucleus, which may be caused by the non-specific staining.

5. As a pH-responsive Cu-doped calcium phosphate nanoparticles, CCPC degradation in the acidic microenvironment may increase calcium ions in the tumor site. Still, the results show that it does not promote the increase of cytoplasmic calcium ions.

6. Previous microneedle studies associated with enzymatic reactions/cell incorporation (especially by Zhen Gu group) are suggested to cite, e.g., PNAS, 112 (27) 8260-8265, 2015.

Reviewer #3 (Remarks to the Author): with expertise in NPs and PDT

This is an interesting paper in which the authors used a microneedle patch to realize the local treatment of tumor. Specifically, the authors chose 5-ALA and CAT for the in situ synthesis of PpIX and production of oxygen. The experiments have been carefully conducted and the results are encouraging. I would like to recommend the publication of the work in NC after minor revision.

Some specific comments:

In the introduction section, some related references regarding the use of 5-ALA for PDT and the GSH depletion-enhanced tumor treatment should be mentioned: Adv. Therap. 2019, 2(5), 1800140. ACS Nano 2021, 15(5), 8039–8068.

The authors used HA for fabricating the microneedle patch. Please discuss the advantages of using

this polymer for the drug loading and release. Could HA be fully degraded in vivo?

The blood biosafety of the patch should be evaluated.

Because the authors used Cu, could this element play a role in chemodynamic therapy? Please discuss.

Please discuss the advantages of this patch by comparing with a hydrogel delivery system that is also loaded with CAT and 5-ALA (and others) as in this work.

Point-by-point response to Reviewers' comments for manuscript:

“Synthetic biology-instructed transdermal microneedle patch for traceable photodynamic therapy”

- Author responses are in **blue color**, all changes in the revised manuscript and Supplementary Information were marked with **yellow color**.
- Specific comments by the reviewers are underlined.

We would like to thank each Reviewer for all constructive comments and questions. In a series of supplied new experiments, we have attempted to address each comment in full, and hope that we were able to do so in a satisfactory fashion. This substantial additional work has undoubtedly further strengthened our manuscript.

Reviewer #1 (Remarks to the Author, with expertise in 5-ALA and PDT therapy)

Comments:

In this manuscript, the authors developed a transdermal theranostic microneedle patch integrated with 5-ALA and catalase (CAT)-coloaded tumor acidity-responsive copper-doped calcium phosphate nanoparticles for efficient 5-ALA-PDT. The preparation and characterizations of MN-CCPCA patch was well arranged and studied. The synergistic effect of oxygen generation, GSH consumption and ion-interference therapy maximized the efficacy of 5-ALA-PDT, which was investigated in several solid tumors.

Response: Thank you very much for your positive comments.

However, the concept of oxygen production and ion-interference enhanced PDT has been previously reported in many literatures, thus this work lacks of sufficient novelty. In addition,

in vivo fluorescence/photoacoustic duplex imaging guided therapy is not supported by solid evidences. The main strength of the work is that the enhanced therapeutic efficacy was systemically studied; however, synergistic antitumor mechanism was not investigated and the results could not fully support the authors' claims. Therefore, I can't recommend publication of this work at NC.

Response: Thank you very much for your comments and we have conducted additional experiments to address the reviewer's concern. According to the reviewer's opinion, we have added the comparison of therapeutic effect in vivo (in both 4T1 and U87MG tumor model) between single irradiation and multiple irradiations guided by FL/PA duplex imaging after application of MN-CCPCA patch in the revised manuscript (**Fig. 7 and Fig. 8**). In addition, we added the molecule mechanism of the HIF-1 α inhibition (**Fig. 3 i-k**), GSH consumption (**Fig. 7g and Fig. 8 h**) and apoptotic enhancement (**Fig. 4f-h, Fig. 7h and Fig. 8 i**) to prove that the co-delivery of 5-ALA and CAT via our theranostic microneedle patch offers a promising strategy to upregulate intratumoral PpIX biosynthesis and optimize treatment effect by leveraging in vivo real-time PpIX and sO₂ imaging.

It should be noted that the highlights of this work are to solve the bottlenecks (the heterogeneity of PpIX metabolism and the limitation of physiological synthesis PpIX concentration) of 5-ALA based photodynamic therapy (5-ALA-PDT) for solid tumors through the design of synthetic biology, optimized delivery strategy and real-time imaging guidance. In our work, the phenomenon and molecular mechanism that molecular oxygen can increase the concentration of intratumoral PpIX are found and a synthetic biology approach is developed to break the physiological limitation of PpIX synthesis in different kinds of solid tumors; Besides, by combining a microneedle transdermal delivery strategy and a sustained-release nano-delivery system for specific/local delivery of 5-ALA and CAT, more synthetic PpIX are trapped in tumor cells instead of synthesizing heme, this combination enhances the bioavailability of 5-ALA and CAT and amplifies the signal of FL/PA duplex imaging simultaneously; Finally, Laser irradiation parameters (time point, dosage, duration, and number of laser irradiation) for different solid tumors (such as breast, glioblastoma, melanoma) can be optimized by FL/PA duplex imaging, which overcomes issues of heterogeneous PpIX metabolism in different tumor types and ultimately improves therapeutic efficacy and safety.

There are also other questions that need to be addressed if this work is going to be published elsewhere.

Major concerns:

1. In the method to measure O₂ generation, PpIX production, and expression of HIF-1 α and FECH in A375 cells placed in incubator at 37 °C with 5% CO₂, it is strange to observe such a difference in differently treated A375 cells. The normoxic environment would also provide enough oxygen level, and affect the detection results. Why did not the authors conduct the experiment in hypoxia environment?

Response: Thank you very much for your valuable comment, we have conducted additional experiments to address the reviewer's concern. According to this valuable suggestion, the additional experiments by immunofluorescence staining (**Fig. 3 j, k**) and western immunoblots analysis (**Fig. 3 l, m**) of HIF-1 α were designed to prove that CCPCA can suppress the expression of HIF-1 α in both normoxia (21% O₂) and hypoxia conditions (1% O₂). HIF-1 α is highly expressed in A375 cells in both normal oxygen and hypoxia conditions (*Mol. Ther.* **17**, 269-277 (2009), *Cancer Res.* **72**, 5035-5047 (2012)), which is consistent with the results of our experiment (**Fig. 3 i-k**). Specifically, oxygen (O₂) is indispensable for the ubiquitination degradation of HIF-1 α (*Science* **292**, 468-472 (2001)) through the stereoelectronic effect (*Angew. Chem., Int. Ed. Engl.* **48**, 1784-1787 (2009)). The reason why we did not perform experiments under hypoxic conditions is that the current high-content analysis systems (HCAS) cannot equip with a real-time hypoxic culture hardware.

Detailed information is described in the section of “**PpIX biosynthesis and outflow shut down**” as follows:

“Importantly, the continuous elevation of endogenous O₂ levels by CAT suppressed the expression of HIF-1 α in both normoxia (21% O₂) and hypoxia conditions (1% O₂, Fig. 3 i-k). The decreased expression of HIF-1 α was observed in a time-dependent (Fig. 3 l) as well as CCPCA concentration-dependent (Fig. 3 m) manner”.

Fig. 3 j Immunofluorescence images and **(k)** quantitative FL intensity analysis of anti- α -tubulin (green) and anti-HIF-1 α (red) in A375 cells after different treatments under hypoxic (1% O₂, 5% CO₂, and 94% N₂) and normoxic (21% O₂, 5% CO₂, and 74% N₂) conditions, respectively. Scale bar = 50 μ m. Western immunoblots analysis of HIF-1 α expression levels in A375 cells treated with CCPCA at different **(l)** time and **(m)** concentrations.

2. In the part of in vivo PpIX metabolism, tumor-bearing mice were treated with MN-CCPCA for only 36 h, and an enhanced production of PpIX metabolism was observed. However, it is a little surprising to see that catalase could alleviate tumor hypoxia microenvironment in tumor-bearing mice and even affected the protein expression (HIF-1 α and FECH) in such a short time (36 h in antitumor study). Details investigations need to be provided.

Response: Thank you very much for your valuable comment, we respectfully disagree with this reviewer on the point of HIF-1 α and FECH inhibition. According to *Nat. Commun.* **7**, 13593 (2016) and *Proc. Natl Acad. Sci. U.S.A.* **92**, 5510–5514 (1995), The expression of HIF-1 α is rapidly induced by hypoxia. When hypoxic cells are reoxygenated, the protein of HIF-1 α will rapidly degrade (half-life of <10 min). This is due to that the specific binding of the von Hippel-Lindau (pVHL) to the C4-exo conformation of the hydroxyprolyl residue results in a >1000-fold increase in affinity of HIF- α for pVHL upon hydroxylation, thus promoting the destruction of HIF-1 α in the presence of O₂ (*Angew. Chem., Int. Ed. Engl.* **48**, 1784-1787 (2009)). Same thing for cancer cells, RCC4, HT1080 and Hela cells were exposed to severe

hypoxia (0.1% O₂ for 4 h) and then re-exposed to ambient oxygen (21% O₂), ubiquitination degradation of HIF-1 α immediately happened within 1 min and reach a new steady-state within a short time (half-life < 60 min), and 3% O₂ was sufficient to cause a significant decrease in the expression of HIF-1 α in Hela cells within 2 h (*J. Biol. Chem.* **286**, 13041-13051 (2011)). Numerous studies (*Adv. Funct. Mater.* **26**, 7847-7860 (2016), *Adv. Funct. Mater.* **31**, 2009848 (2021)) have also shown that precise delivery of CAT-loaded nanoparticles to the tumor site could produce O₂ and reduce the expression of HIF-1 α . In addition, FECH could be significantly inhibited after 20 h administration of HIF-1 α inhibitor under normoxic conditions in human cancer cell lines including DLD-1, SNB-75, and MDA MB 231 cells (*Sci. Rep.* **10**, 22124 (2020)). These results proved that MN-CCPCA could alleviate tumor hypoxia microenvironment in tumor-bearing mice and affect the protein expression of HIF-1 α and FECH within 24 h, which was also consistent with the in vitro (**Fig. 3h and Supplementary Fig. 6d-e**) and in vivo (**Fig.5a-b, Fig.7a-b, and Fig.8a-b**) results of PpIX metabolism dynamics.

3. In the part of oxygen production, catalase catalyzed tumor hydrogen peroxide to produce oxygen. As far as we know, catalase temporarily alleviates hypoxia tumor environment, which has been widely used for enhanced dynamic therapy. However, because the hydrogen peroxide is usually limited in tumor tissues, how did it sustainably produce oxygen? The concentration change of tumor hydrogen peroxide should be also evaluated after treatment with MN-CCPCA. The tumor oxygen level should be measured for at least one week to support the claim of “sustainable oxygen saturation”.

Response: Thank you very much for your valuable comment. Endogenous H₂O₂ sources include NADPH oxidases and other oxidases (membrane-bound or free) as well as the mitochondria (*Redox Biol.* **11**, 613-619 (2017)). The superoxide anion radical is converted to H₂O₂ by the three superoxide dismutases (Superoxide dismutase 1,2,3). It appears from a search of the available literature that H₂O₂ concentration in blood plasma is about 1–5 μ M quantified by genetically encoded fluorescent protein indicators of H₂O₂, which is 650-fold higher than the concentration of intracellular H₂O₂ (*Redox Biol.* **2**, 955-962 (2014)). And H₂O₂ diffusion across membranes occurs by aquaporins (AQP), known as peroxiporins (*J. Biol.*

Chem **282**, 1183-1192 (2007)). Therefore, either of tumor metabolism or tumor blood pools can provide a large amount of H₂O₂ in the tumor microenvironment. As is shown in **Fig 5 d-e**, the sO₂ level after MN-CCPC treatment at 1, 24, 48 h was 14.6%, 21.9%, 24.5%, respectively, while the sO₂ level after MN-CCP treatment at 1, 24, 48 h was 8.1%, 12.6%, 12.6%, respectively, indicating that tumor sO₂ level can be increased within 48 h by loading CAT. Because MN-CCPCA could alleviate tumor hypoxia microenvironment in tumor-bearing mice and affected the protein expression of HIF-1 α and FECH within 24 h (already explained in the answer of third question), we believe that the improved sO₂ level within 48 hours can provide enough time window for the diagnosis and treatment of solid tumors.

Fig. 5 d Representative PA images of sO₂ in A375 tumors before and after applying MN-CCP, MN-CCPC, MN-CCPA and MN-CCPCA patches at a given time interval. All the groups were irradiated with a 635 nm laser (200 mW cm⁻², 10 min) after testing the sO₂ at the 24 h time point. **e** Quantitative analysis of the sO₂ in the total tumor area at different time points. The data represent the means \pm SD, (n = 3).

4. In the antitumor treatment, why were the mice irradiated at 6, 12 and 24 h post-transdermal application of patches. Is there any evidence to support the irradiation for three times, as the tumor oxygen level could be reduced after each PDT.

Response: Thank you very much for your valuable comment. PpIX and O₂ are essential substrates for the production of ¹O₂ during laser irradiation. On one hand, maximizing the enrichment of intratumoral PpIX is a prerequisite for 5-ALA-PDT (*Adv. Mater.* **32**, 2004481 (2020), *Adv. Mater.* **32**, 2004481 (2020)). As the heterogeneity of solid tumors, the maximum

enrichment time point (METP) of PpIX in the A375, 4T1 and U87MG tumor models were recorded at 12, 24, and 12 h, respectively, so we firstly choose the METP for laser irradiation. On the other hand, because intermittent irradiation during PDT can alleviate hypoxia (*Adv. Mater.* **31**, 1900927 (2019), *ACS Nano* **14**, 15793-15805 (2020)), laser irradiation at the point when the maximum PpIX concentration accumulates in the tumors combined with a long irradiation interval will facilitate the restoration of intratumoral O₂ levels, so the U87MG and A375 tumor-bearing mice were irradiated at 6, 12 and 24 h, while the 4T1 tumor-bearing mice were irradiated at 12, 24 and 36 h (Here, 3 times irradiation was only used as an example of multiple irradiations).

To further prove that repeated PDT is more efficient than single irradiation at the same dosage of laser power, an additional experimental group of MN-CCPCA (S), in which the MN-CCPCA-treated 4T1 tumor-bearing mice received a single irradiation (200 mW cm⁻², 30 min) at the METP of PpIX, was used as a contrast to assess the enhanced therapeutic efficacy of repeated PDT (**Fig. 7 c-h and Supplementary Fig. 11a-g**). Detailed information is described in the section of “**MN administration Improved the bioavailability and therapeutic effect in vivo**” as follows:

“We selected 12, 24, and 36 h as the multiple irradiation time (200 mW cm⁻², 10 min each time) for the MN-CCPCA (M) group (Fig. 7c). The other treatment groups received a single irradiation (200 mW cm⁻², 30 min) at the METP of PpIX according to its metabolic kinetic analysis (Fig. 7a-b). The 4T1 tur-bearing mice were randomly divided into 7 groups: i) Control (S), ii) 5-ALA *i.v.* (S), iii) 5-ALA *i.t.* (S), iv) CCPCA *i.v.* (S), v) CCPCA *i.t.* (S), vi) MN-CCPCA (S) and vii) MN-CCPCA (M). The 4T1 tumor growth inhibition rates of 5-ALA *i.t.* (S), MN-5-ALA (S), CCPCA *i.t.* (S), MN-CCPCA (S) and MN-CCPCA (M) groups were calculated as 47.4%, 51.1%, 59.8%, 78.7% and 92.3%, respectively (Fig. 7c-d). The low 5-ALA bioavailability and PpIX synthesis efficiency in the 5-ALA *i.v.* (S) as evidenced by the metabolic FL images of PpIX (Fig. 7a and b) led to a poor therapeutic effect (Fig. 7c-f). Similarly, tumor recurrence occurred in all 4T1 tumor-bearing mice in 5-ALA *i.t.* (S), MN-5-ALA (S), CCPCA *i.t.* (S) and MN-CCPCA (S) groups (Fig. 7c-f). Nevertheless, the MN-CCPCA (M) group showed dramatic tumor growth inhibition within 2 weeks, indicating that intermittent irradiation in MN-CCPCA group can reduce the recurrence rate of 4T1 tumors. In addition,

negligible change of the relative body weight of 4T1 tumor-bearing mice (Supplementary Fig. 11a) also proved that the optimized treatment strategy based on the MN-CCPCA patch has good biosafety.

Fig. 7 Antitumor effects of MN-CCPCA patch in a 4T1 tumor model. **c** Individual tumor growth curves, **(d)** relative tumor volume growth curves and **(e)** 4T1 tumor weight in different treatment groups on day 14, the symbols “S” indicate single laser irradiation and “M” indicate multiple laser irradiation. **f** Photographs of excised 4T1 tumors on day 14 after different treatments. **g** In vivo immunofluorescence images of anti-GSH (green) and Hoechst (blue) in 4T1 tumors after different treatments; scale bar = 1 mm. **h** Representative images of HIF-1 α -, H&E-, TUNEL-stained sections of 4T1 tumors after various treatments; scale bar = 100 μm . The symbols “+” indicate laser irradiation. The data represent the means \pm SD, (n = 6). ***p < 0.001, **p < 0.01, *p < 0.05, n.s., not significant. Two-tailed Student’s t test.

Supplementary Fig. 11 Antitumor effects of MN-CCPCA patch in a 4T1 tumor model. (a) Body weight of 4T1 tumor-bearing mice in different treatment groups, (n=6). Accurate quantitative of **(b)** tumor area, **(c)** positive area ratio and **(d)** positive area density of anti-GSH FL signals in 4T1 tumors by Aipathwell immunofluorescence analysis. Quantitative analysis of **(e)** HIF-1α and **(f)** dUDP FL signals in 4T1 tumors. **(g)** Ki67-stained sections of 4T1 tumors after various treatments. The data represent the means \pm SD, (n = 5), ***p < 0.001, **p < 0.01, *p < 0.05, n.s., not significant. Statistical differences were determined by Two-tailed Student's t test.

Importantly, the in vivo immunofluorescence images of anti-GSH level showed that all the treatment groups containing CCPCA could decrease the GSH levels, the positive area ratio and positive area density of anti-GSH in 4T1 tumor treated with MN-CCPCA (M) were calculated as 0.31% and 0.0003, respectively (Supplementary Fig. 11b-d). Besides, the HIF-1 α expression levels in the CAT-loaded groups were lower than in the other groups as the consumption of O₂ by 5-ALA-PDT without CAT caused more severe hypoxia (which may result in resistance to PDT in 4T1 tumors). It's worth noting that a higher expression level of HIF-1 α in the MN-CCPCA (S) group was observed than the MN-CCPCA (M) group, indicating that intermittent irradiation is helpful to reduce tumor hypoxia (Fig. 7h and Supplementary Fig. 11e). By comparing H&E, TUNEL and Ki67 staining images (Fig. 7h and Supplementary Fig. 11f-g), the nuclei in the MN-CCPCA (M) group exhibited most severe karyopyknosis, strongest apoptotic signal and negligible proliferative signal. These results confirmed that the MN-CCPCA patch has efficient antitumor effects in a 4T1 tumor model when combining a microneedle transdermal delivery strategy with adjustable irradiation parameters under the "FL/PA duplex imaging guidance".

5. An enhanced antitumor therapeutic efficacy of MN-CCPCA was detected in different tumor models (breast cancer, glioblastoma, melanoma). The authors declared the O₂ generation by PpIX along with Cu²⁺/Ca²⁺ released by CCPCA NPs not only consumed antioxidant glutathione but also achieved stronger cytosolic free Ca²⁺ ([Ca]_{CYT}) overload for ion-interference enhanced repeatable 5-ALA-PDT. However, only antitumor result was shown in this study, and there was no mechanism study conducted to support this claim at molecular level.

Response: Thank you very much for your valuable comment. To support the GSH depletion ability of the MN-CCPCA patch, in vivo immunofluorescence imaging of anti-GSH was investigated in vivo both in 4T1 and U87MG tumors (**Fig. 7g and Fig. 8h**), and our results showed that all the treatment groups containing CCPCA could deplete the GSH, resulting in low tumor area, positive area ratio and positive area density of GSH FL signals (**Supplementary Fig. 11b-d and Supplementary Fig. 13c-e**).

Fig. 7 g In vivo immunofluorescence imaging of anti-GSH (green) and Hoechst (blue) in 4T1 tumors after different treatments; scale bar = 1 mm.

Supplementary Fig. 11 Accurate quantitative of (b) tumor area, (c) positive area ratio and (d) positive area density of anti-GSH FL signals in 4T1 tumors by Aipathwell immunofluorescence analysis.

Fig. 8 h In vivo immunofluorescence imaging of anti-GSH (green) and Hoechst (blue) in U87MG tumors after different treatments; scale bar = 1 mm.

Supplementary Fig. 13 Accurate quantitative of (b) tumor area, (c) positive area ratio and (d) positive area density of anti-GSH FL signals in U87MG tumors by Aipathwell immunofluorescence analysis.

To support the HIF-1 inhibition ability of CCPCA, immunofluorescence staining and western immunoblots analysis of HIF-1 α were designed to prove that CCPCA can suppressed the expression of HIF-1 α in vitro in both normoxia (21% O₂) and hypoxia conditions (1% O₂, already explained in the answer of third question). In vivo HIF-1 α immunofluorescence staining images of A375, 4T1 and U87MG tumors (Fig. 6f, Fig. 7h and Fig. 8i) all confirmed that the expression of HIF-1 α was successfully inhibited by the MN-CCPCA patch.

Fig. 7 h Representative images of HIF-1 α -stained sections of 4T1 tumors after various treatments, scale bar = 100 μ m.

Supplementary Fig. 11 e Quantitative analysis of HIF-1 α signals in 4T1 tumors after various treatments.

Fig. 8 i Representative images of HIF-1 α -stained sections of U87MG tumors after various treatments, scale bar = 100 μ m.

Supplementary Fig. 13 e Quantitative analysis of HIF-1 α signals in U87MG tumors after various treatments.

To demonstrate that the delivery of 5-ALA and CAT using CCPCA nanoplatform could not only reduce HIF-1 α , but also enhance the apoptotic signal caused by calcium overload, immunofluorescence imaging and corresponding quantitative FL intensity analysis of anti-caspase-3 (green) and anti-HIF-1 α (red) in A375 cells after different treatments under hypoxic (1% O₂, 5% CO₂, and 94% N₂) conditions were investigated in **Fig. 4f-h**, which are described in detail in the section of “**Synergistic mechanism of ion interference-enhanced 5-ALA-PDT**” as follows:

“Similarly, after irradiation at hypoxia condition for 5 min, a significant enhancement of the FL intensity of caspase-3 in the CCPCA group (193.6%) was observed compared with the 5-ALA (111.3%) and CCPA (151.4%) groups (Fig. 4g), while the expression of HIF-1 α in the CCPCA (+) group was only 26.4% of the control (+) group (Fig. 4h), indicating that CCPCA can improve the phototoxicity both in normoxia and hypoxia conditions”.

Fig. 4 **f** Immunofluorescence images and corresponding quantitative FL intensity analysis of **(g)** anti-caspase-3 (green) and **(h)** anti-HIF-1α (red) in A375 cells after different treatments under hypoxic (1% O₂, 5% CO₂, and 94% N₂) conditions. A375 cells were incubated with blank DMEM, 5-ALA, CCPA and CCPCA NPs for 24 h and then irradiation with a 635 nm laser (200 mW cm⁻², 5 min), scale bar = 50 μm. ***p < 0.001, **p < 0.01, n.s., not significant. One-way ANOVA with Tukey's multiple comparisons test.

6. In Fig. 7 and Fig. 8, 12, 24, and 36 h were selected as the irradiation times (200 mW cm⁻², 10 min each time) for the MN-CCPCA (+) group, while the other treatment groups were irradiated for only one time at the METP of PpIX according to its metabolic kinetic analysis. The comparison of therapy efficacy here would not be fair.

Response: We thank the reviewer for the constructive suggestion on comparison of therapy efficacy. It needs to be emphasized that the power of 3 times irradiation of 200 mW cm⁻² for 10 min and single irradiation of 200 mW cm⁻² for 30 min are the same. We agree with the reviewer that there is a lack of a single irradiation group (200 mW cm⁻² for 30 min) of MN-CCPCA (MN-CCPCA (S)). Both 4T1 (already explained in the answer of third question) and U87MG tumor models added this treatment group. Detailed information of therapeutic effect of MN-CCPCA (S) group in U87MG tumor model (**Fig. 8 d-j** and **Supplementary Fig. 13a-i**) is described in the section of **"In vivo personalized theranostics for the U87MG tumor model"** as follows:

"The bioluminescence images of U87MG tumor-bearing mice before (day 0) and after treatment (day 14) suggested that all these administration modes of 5-ALA showed a certain

tumor growth inhibition after laser irradiation (Fig. 8d, and Supplementary Fig. 13a-b). The U87MG tumor growth inhibition rates of 5-ALA *i.t.* (S), MN-5-ALA (S), CCPCA *i.t.* (S), MN-CCPCA (S) and MN-CCPCA (M) groups were calculated as 60.8%, 79.9%, 82.4%, 88.3% and 95.2%, respectively (Fig. 8f, g). As is the case in the 4T1 model, tumor recurrence occurred in both *i.v.* and *i.t.* paths of 5-ALA or CCPCA groups (Fig. 8d-f and Supplementary Fig. 13a-b). However, the MN-CCPCA (S) and the MN-CCPCA (M) groups did not show tumor recurrence within 2 weeks and even displayed complete tumor elimination in U87MG tumors (Fig. 8d-f and Supplementary Fig. 13b, c).

Consistent with our previous results of 4T1 tumor sections, we found that groups treated with CCPCA could deplete the GSH, resulting in small tumor area, low positive area ratio and positive area density of anti-GSH FL signals in U87MG tumors (Fig. 8h and Supplementary Fig. 13c-e), and the hypoxic microenvironment in U87MG tumors was significantly reversed in groups contained CCPCA according to HIF-1 α staining images (Fig. 8i and Supplementary Fig. 13f). Compared to the lower local apoptosis FL signal of fluorescein-dUTP in other groups, the higher fully covered FL signal of fluorescein-dUTP in the MN-CCPCA (M) group also revealed the most intense signal of apoptosis in tumors (Fig. 8i and Supplementary Fig. 13g). Severely pyknotic nuclei and negligible Ki67 expression were observed in MN-CCPCA (M) groups (Fig. 8i and Supplementary Fig. 13h, i).

As we know, PpIX can be activated by UV or visible light, causing systemic porphyria and damage to normal tissues⁷. The uncontrolled release of 5-ALA and nonspecific distribution of PpIX by *i.v.* or *i.t.* injections of CCPCA raised biosafety concerns for following irradiation treatment. Although there was no evident damage to major organs of U87MG tumor-bearing mice after *i.v.*, *i.t.* or MN administrations of CCPCA within 14 days (Supplementary Fig. 14), we further conducted serum biochemical analysis to assess the biosafety of *i.t.* or MN administrations of CCPCA before and after laser irradiation. As shown in Fig 8j, there was no significant difference in serum biochemical parameters without laser exposure. However, alanine aminotransferase (ALT) and aspartate aminotransferase (AST) significantly increased in CCPCA *i.t.* group after 30 min laser irradiation. In contrast, there was negligible difference in serum biochemical parameters of liver and kidney function of mice treated with

MN-CCPCA patch after 30 min laser irradiation, suggesting that MN administration of CCPCA can improve the biosafety of 5-ALA-PDT”.

Fig. 8. Enhanced therapeutic efficacy and biosafety of MN-CCPCA patch for U87MG tumors. **d** Bioluminescence images of U87MG tumor-bearing mice treated with 5-ALA *i.v.*, 5-ALA *i.t.*, MN-5-ALA, CCPCA *i.t.*, MN-CCPCA (S), MN-CCPCA (M) on days 0 and 14. The symbols “S” indicate single laser irradiation and “M” indicate multiple laser irradiation. **e** Relative tumor volume growth curves, **(f)** tumor weight and **(g)** body weight in different treatment groups on day 14, (n = 5). **h** In vivo immunofluorescence images of anti-GSH (green) and Hoechst (blue) in 4T1 tumors after different treatments; scale bar = 1 mm. **i**

Representative images of HIF-1 α -, H&E-, and TUNEL-stained sections of U87MG tumors after various treatments, scale bar = 100 μ m. j Serum biochemical analysis of U87MG tumor-bearing mice in CCPCA *i.t.* and MN-CCPCA groups before and after 30 min laser irradiation, (n = 3). The data represent the means \pm SD. ***p < 0.001, **p < 0.01, *p < 0.05, n.s., not significant. Two-tailed Student's t test.

Supplementary Fig. 13 Antitumor effects of MN-CCPCA patch in a 4T1 tumor model. (a)

Photographs of U87MG tumor-bearing mice at the end of treatment (day 14). **(b)** Individual tumor growth curves of U87MG tumor weight in different treatment groups on day 14. Accurate quantitative of **(c)** tumor area, **(d)** positive area ratio and **(E)** positive area density of anti-GSH FL signals in U87MG tumors by Aipathwell immunofluorescence analysis. Quantitative analysis of **(f)** HIF-1 α , **(g)** dUDP and **(h)** Ki67 FL intensity in U87MG tumor slices after different treatments. **(i)** Ki67-stained sections of U87MG tumors after various treatments. The data represent the means \pm SD, (n = 5), ***p < 0.001, **p < 0.01, *p < 0.05, n.s., not significant. Statistical differences were determined by Two-tailed Student's t test.

7. In Fig. 5d and 5e, MN-CCP and MN-CCPA should be studied.

Response: Thank you very much for your valuable comment, we have added the study of MN-CCP and MN-CCPA groups in the revised manuscript. The sO₂ levels of MN-CCP and MN-CCPA groups by PA imaging has been described in detail in the section of **“Monitoring PpIX biosynthesis in vivo by real-time FL/PA duplex imaging”** as follows:

“Even so, the level of sO₂ in the MN-CCPCA group was 1.68 and 2.49 times than that of the control and MN-CCPA groups at 24 h post-irradiation, respectively, and suggesting that the MN-CCPCA patch could relieve tumor hypoxia during intermittent irradiation (Fig. 5e)”.

Fig. 5 d Representative PA images of sO₂ in A375 tumors before and after applying MN-CCP, MN-CCPC, MN-CCPA and MN-CCPCA patches at a given time interval. All the groups were irradiated with a 635 nm laser (200 mW cm⁻², 10 min) after testing the sO₂ at the 24 h time point. **e** Quantitative analysis of the sO₂ in the total tumor area at different time points. The data represent the means \pm SD, (n = 3).

8. In line 50-51, “a short treatment window and a short treatment window” should be revised, etc.

Response: Thank you very much for your valuable comment. We have deleted the repetition part of “and a short treatment window” in the revised manuscript.

Reviewer #2 (Remarks to the Author, with expertise in microneedles and NPs)

Comments:

In this manuscript, Huang and coworkers leveraged a synthetic biology approach to construct a transdermal theranostic microneedle patch integrated with 5-Aminolevulinic acid (5-ALA) and catalase (CAT)-coloaded calcium phosphate nanoparticles for efficient 5-ALA-PDT. The author demonstrated that the continuous oxygen generation by CAT *in vivo* reverses tumor hypoxia and could promote protoporphyrin IX (PpIX) accumulation. Besides that, the *in vivo* fluorescence/photoacoustic duplex imaging could monitor intratumoral oxygen saturation and PpIX metabolic kinetics simultaneously.

Response: Thank you very much for your positive comment.

However, the advantage of delivery routing by microneedle compared with intratumor injection in this study is unclear. The concept of "Synthetic biology-instructed..." seems oversold here. Overall, this manuscript needs a major revision before potential acceptance.

1. The authors need to clarify the advantage and necessity of a microneedle patch for the nano-mixture delivery in this study, and another control group, intratumor injection of the therapeutic agents, should be added.

Response: Thank you very much for your constructive comment. According to this valuable suggestion, we firstly designed fluorescence (FL) imaging experiments in both 4T1 and U87MG tumor-bearing mice models to demonstrate the improved bioavailability and biosafety of MN-CCPCA administration comparing to other delivery modes (intravenous injection (*i.v.*), intratumor injection (*i.t.*)) of 5-ALA or CCPCA.

The biological distributions of PpIX in 4T1 tumor model by *i.v.*, *i.t.* and MN administration of 5-ALA and CCPCA were determined by FL imaging (**Fig. 7 a-b and Supplementary Fig. 10a-d**) and corresponding therapeutic effect has also been investigated (**Fig. 7 c-h and Supplementary Fig. 11f-g**). which are described in detail in the section of "**MN administration Improved the bioavailability and therapeutic effect *in vivo***" as follows:

"To confirm whether our developed theranostic microneedle patch (MN-CCPCA) could effectively treat solid malignant tumors by optimizing the delivery strategies and complex

parameters of 5-ALA-PDT, the biological distributions of PpIX after intravenous injection (*i.v.*), intratumoral (*i.t.*) injection or MN administration of 5-ALA and CCPCA were determined by FL imaging, respectively. As the high interstitial fluid pressure of 4T1 solid tumors⁴⁹, negligible FL intensity of PpIX was accumulated at tumor sites after *i.v.* administration of 5-ALA or CCPCA (Fig. 7a and b). The METPs of intratumoral PpIX in the 5-ALA *i.t.*, CCPCA *i.t.*, MN-5-ALA, and MN-CCPCA groups were 1, 12, 12 and 24 h, respectively (Fig. 7a and b). The 2.3- and 4.7-fold higher FL intensities of PpIX in the MN-CCPCA group were recorded at METP (24 h) than those of CCPCA *i.t.* and MN-5-ALA groups, respectively. Both *in vivo* and *ex vivo* FL images of 4T1 tumor-bearing mice indicated that a higher PpIX concentration and longer tumor retention of PpIX can be achieved in 4T1 tumors by MN-CCPCA treatment (Supplementary Fig. 10 a-d).

Because intermittent irradiation during PDT can alleviate hypoxia^{50, 51}, laser irradiation at the point when the maximum PpIX concentration accumulates in the tumors combined with a long irradiation interval (intermittent irradiation) will facilitate the restoration of intratumoral O₂ levels, resulting in high ¹O₂ generation to realize efficient antitumor effects. Based on these facts, we selected 12, 24, and 36 h as the multiple irradiation time (200 mW cm⁻², 10 min each time) for the MN-CCPCA (M) group (Fig. 7c). The other treatment groups received a single irradiation (200 mW cm⁻², 30 min) at the METP of PpIX according to its metabolic kinetic analysis (Fig. 7a-b). The 4T1 tumor-bearing mice were randomly divided into 7 groups: i) Control (S), ii) 5-ALA *i.v.* (S), iii) 5-ALA *i.t.* (S), iv) CCPCA *i.v.* (S), v) CCPCA *i.t.* (S), vi) MN-CCPCA (S) and vii) MN-CCPCA (M). The 4T1 tumor growth inhibition rates of 5-ALA *i.t.* (S), MN-5-ALA (S), CCPCA *i.t.* (S), MN-CCPCA (S) and MN-CCPCA (M) groups were calculated as 47.4%, 51.1%, 59.8%, 78.7% and 92.3%, respectively (Fig. 7c-d). The low 5-ALA bioavailability and PpIX synthesis efficiency in the 5-ALA *i.v.* (S) as evidenced by the metabolic FL images of PpIX (Fig. 7a and b) led to a poor therapeutic effect (Fig. 7c-f). Similarly, tumor recurrence occurred in all 4T1 tumor-bearing mice in 5-ALA *i.t.* (S), MN-5-ALA (S), CCPCA *i.t.* (S) and MN-CCPCA (S) groups (Fig. 7c-f). Nevertheless, the MN-CCPCA (M) group showed dramatic tumor growth inhibition within 2 weeks, indicating that intermittent irradiation in MN-CCPCA group can reduce the recurrence rate of 4T1 tumors. In addition, negligible change of the relative body weight of 4T1 tumor-bearing mice (Supplementary Fig.

11a) also proved that the optimized treatment strategy based on the MN-CCPCA patch has good biosafety.

Fig. 7 Antitumor effects of MN-CCPCA patch in a 4T1 tumor model. **a** In vivo PpIX FL images of 4T1 tumor-bearing mice after administration of 5-ALA *i.v.*, 5-ALA *i.t.*, MN-5-ALA, CCPCA *i.v.*, CCPCA *i.t.*, and MN-CCPCA patch at the indicated time points. **b** PpIX FL intensity of tumor tissues after different treatments at the indicated time points. **c** Individual tumor growth curves, **(d)** relative tumor volume growth curves and **(e)** 4T1 tumor weight in

different treatment groups on day 14, the symbols “S” indicate single laser irradiation and “M” indicate multiple laser irradiation. **f** Photographs of excised 4T1 tumors on day 14 after exposure to different treatments. **g** In vivo immunofluorescence imaging of anti-GSH (green) and Hoechst (blue) in 4T1 tumors after different treatments; scale bar = 1 mm. **h** Representative images of HIF-1 α -, H&E-, TUNEL-stained sections of 4T1 tumors after various treatments; scale bar = 100 μ m. The data represent the means \pm SD, (n = 6). ***p < 0.001, **p < 0.01, *p < 0.05, n.s., not significant. Two-tailed Student’s t test.

Supplementary Fig. 10 Ex vivo FL imaging of major organs and tumors in 4T1 tumor model after different administrations of 5-ALA and CCPCA. (a) FL images of major organs and tumors of 4T1 tumor-bearing mice and quantification of FL intensity of PpIX after application of **(b)** 5-ALA *i.v.* and CCPCA *i.v.*, **(c)** 5-ALA *i.t.* and CCPCA *i.t.*, **(d)** MN-5-ALA and MN-CCPCA in major organs and tumors ex vivo. Data are presented as mean \pm SD, n = 3.

Supplementary Fig. 11 Antitumor effects of MN-CCPCA patch in a 4T1 tumor model. (a) Body weight of 4T1 tumor-bearing mice in different treatment groups, (n=6). Accurate quantitative of **(b)** tumor area, **(c)** positive area ratio and **(d)** positive area density of anti-GSH FL signals in 4T1 tumors by Aipathwell immunofluorescence analysis. Quantitative analysis of **(e)** HIF-1α and **(f)** dUDP FL signals in 4T1 tumors. **(g)** Ki67-stained sections of 4T1 tumors after various treatments. The data represent the means ± SD, (n = 5), ***p < 0.001, **p < 0.01, *p < 0.05, n.s., not significant. Statistical differences were determined by Two-tailed Student's t test.

Importantly, the in vivo immunofluorescence images of anti-GSH level showed that all the treatment groups containing CCPCA could decrease the GSH levels, the positive area ratio and positive area density of anti-GSH in 4T1 tumor treated with MN-CCPCA (M) were calculated as 0.31% and 0.0003, respectively (Supplementary Fig. 11b-d). Besides, the HIF-1α expression levels in the CAT-loaded groups were lower than in the other groups as the consumption of O₂ by 5-ALA-PDT without CAT caused more severe hypoxia (which may

result in resistance to PDT in 4T1 tumors). It's worth noting that a higher expression level of HIF-1 α in the MN-CCPCA (S) group was observed than the MN-CCPCA (M) group, indicating that intermittent irradiation is helpful to reduce tumor hypoxia (Fig. 7h and Supplementary Fig. 11e). By comparing H&E, TUNEL and Ki67 staining images (Fig. 7h and Supplementary Fig. 11f-g), the nuclei in the MN-CCPCA (M) group exhibited most severe karyopyknosis, strongest apoptotic signal and negligible proliferative signal. These results confirmed that the MN-CCPCA patch has efficient antitumor effects in a 4T1 tumor model when combining a microneedle transdermal delivery strategy with adjustable irradiation parameters under the FL/PA duplex imaging guidance”.

Similarly, the biological distributions of PpIX after different administration methods in U87MG tumor-bearing mice were determined by FL imaging (**Fig.8 a-c and Supplementary Fig. 12a-e**), and corresponding therapeutic effect has also been investigated (**Fig.8 d-j and Supplementary Fig. 13a-i**). which are described in detail in the section of “**In vivo personalized theranostics for the U87MG tumor model**” as follows:

“We saw that PpIX was synthesized throughout the body after *i.v.* administration of 5-ALA or CCPCA, especially in the brain (Fig. 8a-b), which may cause organelle-selective protein aggregation and phototoxicity to normal tissues as a higher concentration of PpIX than physiological condition^{7, 26, 52}. Obvious FL signals of PpIX in both brain and normal skin tissues around tumors were also observed after *i.t.* administration of 5-ALA or CCPCA (Fig. 8a-b). The METP of MN-5-ALA group was recorded at 6 h post-administration, then the FL signals of PpIX rapidly cleared from the tumors, indicating short retention time of synthetic PpIX in tumor tissues. The partial amount of PpIX was nonspecifically distributed throughout the whole body due to the uncontrolled release of 5-ALA (Fig. 8a and Supplementary Fig. 12a-d). By contrast, PpIX was almost synthesized at the tumor sites and displayed a longer tumor retention time in the MN-CCPCA group (Fig. 8a-b and Supplementary Fig. 12a-d), as 6.87- and 5.73-fold higher FL intensities of PpIX were recorded at 24 h than those of the CCPCA *i.t.* and MN-5-ALA groups, respectively (Fig. 8 a-b). In addition, the infiltration depth of PpIX in the MN-CCPCA group was more than 1.2 mm along the pierced direction of the tips with 1.96-fold and 1.24-fold increase of PpIX FL intensities across the skin than MN-5-

ALA and CCPCA *i.t.* groups at 12 h, respectively. (Fig. 8c and Supplementary Fig.12e). These results supported that MN-CCPCA can deliver 5-ALA in a controlled manner with improved bioavailability than the other delivery modes (*i.v.*, *i.t.*) of 5-ALA and CCPCA, as we also observed in a 4T1 breast tumor model (Fig. 7a, b and Supplementary Fig.10a-d). Nevertheless, the METPs of PpIX in the A375, 4T1 and U87MG tumor models were recorded at 12, 24, and 12 h, respectively (Fig. 5a, Fig. 7a and Fig. 8a), indicating that the heterogeneous metabolic dynamics of PpIX in different types of tumors are attributed to their intrinsic heterogeneity.

We next irradiated U87MG tumor-bearing mice with a 635 nm laser at 6, 12, and 24 h after the transdermal application of the MN-CCPCA patch according to the metabolic FL images of PpIX. The bioluminescence images of U87MG tumor-bearing mice before (day 0) and after treatment (day 14) suggested that all these administration modes of 5-ALA showed a certain tumor growth inhibition after laser irradiation (Fig. 8d, and Supplementary Fig. 13a-b). The U87MG tumor growth inhibition rates of 5-ALA *i.t.* (S), MN-5-ALA (S), CCPCA *i.t.* (S), MN-CCPCA (S) and MN-CCPCA (M) groups were calculated as 60.8%, 79.9%, 82.4%, 88.3% and 95.2%, respectively (Fig. 8f, g). As is the case in the 4T1 model, tumor recurrence occurred in both *i.v.* and *i.t.* paths of 5-ALA or CCPCA groups (Fig. 8d-f and Supplementary Fig. 13a-b). However, the MN-CCPCA (S) and the MN-CCPCA (M) groups did not show tumor recurrence within 2 weeks and even displayed complete tumor elimination in U87MG tumors (Fig. 8d-f and Supplementary Fig. 13a, b).

Consistent with our previous results of 4T1 tumor sections, we found that groups treated with CCPCA could deplete the GSH, resulting in small tumor area, low positive area ratio and positive area density of anti-GSH FL signals in U87MG tumors (Fig. 8h and Supplementary Fig. 13c-e), and the hypoxic microenvironment in U87MG tumors was significantly reversed in the treatment groups contained CCPCA according to HIF-1 α staining images (Fig. 8i and Supplementary Fig. 13f). Compared to the lower local apoptosis FL signal of fluorescein-dUTP in other groups, the higher fully covered FL signal of fluorescein-dUTP in the MN-CCPCA (M) group also revealed the most intense signal of apoptosis in tumors (Fig. 8i and Supplementary Fig. 13g). Severely pyknotic nuclei and negligible Ki67 expression were observed in MN-CCPCA (M) groups (Fig. 8i and Supplementary Fig. 13h, i).

Fig. 8. Enhanced therapeutic efficacy and biosafety of MN-CCPCA patch for U87MG

tumors. a In vivo PpIX FL images of U87MG tumor-bearing mice after administration of 5-ALA *i.v.*, 5-ALA *i.t.*, MN-5-ALA, CCPCA *i.v.*, CCPCA *i.t.*, and MN-CCPCA patch at the indicated time points. **b** PpIX FL intensity of tumor tissues after different treatments at the indicated time points. **c** Colocalization FL images of intratumoral PpIX by cross sections after different treatments for 12 h. (red from PpIX and blue from Hoechst), scale bar = 200 μm . **d** Bioluminescence images of U87MG tumor-bearing mice treated with 5-ALA *i.v.*, 5-ALA *i.t.*, MN-5-ALA, CCPCA *i.t.*, MN-CCPCA (S), MN-CCPCA (M) on days 0 and 14. The symbols “S” indicate single laser irradiation and “M” indicate multiple laser irradiation. **e** Relative tumor volume growth curves (f) tumor weight and (g) body weight in different treatment groups on day 14, (n = 5). **h** In vivo immunofluorescence imaging of anti-GSH (green) and Hoechst (blue) in 4T1 tumors after different treatments; scale bar = 1 mm. **i** Representative images of HIF-1 α -, H&E-, and TUNEL-stained sections of U87MG tumors after various treatments, scale bar = 100 μm . **j** Serum biochemical analysis of U87MG tumor-bearing mice in CCPCA *i.t.* and MN-CCPCA treatment groups before and after 30 min laser irradiation, (n = 3). The data represent the means \pm SD. ***p < 0.001, **p < 0.01, *p < 0.05, n.s., not significant. Two-tailed Student’s t test.

Supplementary Fig. 12 Ex vivo FL imaging of major organs and tumors in U87MG tumor

model after different administrations of 5-ALA and CCPCA. Quantification of FL intensity of PpIX after application of **(a)** 5-ALA *i.v.* and CCPCA *i.v.*, **(b)** 5-ALA *i.t.* and CCPCA *i.t.*, **(c)** MN-5-ALA and MN-CCPCA, and **(d)** corresponding FL image in major organs and tumors *ex vivo*, (n = 3). **(e)** Quantitative analysis of PpIX FL signals in 4T1 tumors after various treatments. Data are presented as mean \pm SD, (n = 5). Statistical differences were determined by Two-tailed Student's t test.

Supplementary Fig. 13 Antitumor effects of MN-CCPCA patch in a U87MG tumor model.

(a) Photographs of U87MG tumor-bearing mice at the end of treatment (day 14). (b) Individual tumor growth curves of U87MG tumor weight in different treatment groups on day 14. Accurate quantitative of (c) tumor area, (d) positive area ratio and (e) positive area density of anti-GSH FL signals in U87MG tumors by Aipathwell immunofluorescence analysis. Quantitative analysis of (f) HIF-1 α , (g) dUDP and (h) Ki67 FL intensity in U87MG slices after different treatment. (i) Ki67-stained sections of U87MG tumors after various treatments. The data represent the means \pm SD, (n = 5), ***p < 0.001, **p < 0.01, *p < 0.05, n.s., not significant. Statistical differences were determined by Two-tailed Student's t test.

As we know, PpIX can be activated by UV or visible light, causing systemic porphyria and damage to normal tissues⁷. The uncontrolled release of 5-ALA and nonspecific distribution of PpIX by *i.v.* or *i.t.* injections of CCPCA raised biosafety concerns for following irradiation treatment. Although there was no evident damage to major organs of U87MG tumor-bearing mice after *i.v.*, *i.t.* or MN administrations of CCPCA within 14 days (Supplementary Fig. 14), we further conducted serum biochemical analysis to assess the biosafety of *i.t.* or MN administrations of CCPCA before and after laser irradiation. As shown in Fig 8j, there was no significant difference in serum biochemical parameters without laser exposure. However, alanine aminotransferase (ALT) and aspartate aminotransferase (AST) significantly increased in CCPCA *i.t.* group after 30 min laser irradiation. In contrast, there was negligible difference in serum biochemical parameters of liver and kidney function of mice treated with MN-CCPCA patch after 30 min laser irradiation, suggesting that MN administration of CCPCA can improve the biosafety of 5-ALA-PDT”.

2. In this article, the authors demonstrate that Cu produces more reactive oxygen species and enhances tumor killing by consuming glutathione, but this conflicts with the role of CAT, which may lead to higher levels of reactive oxygen species and inhibit the production of PpIX. The authors need more evidence to explain the internal relationship between them.

Response: Thank you very much for your comment. It should be noted that we did not mention that Cu produces reactive oxygen species (ROS) in the manuscript. In contrast, we demonstrated that CAT within CCPCA catalyzes H₂O₂ much more efficiently than Cu²⁺ and

Cu⁺ in the presence of equal amounts of CAT and Cu (The weight percentages of CAT and Cu in CCPCA are 7.9 wt.% and 6.3 wt.%, respectively. And here we assume that CAT and Cu²⁺ in CCPCA are completely released, resulting in a weight ratio close to 1:1), which leads to the inability of Cu²⁺ and Cu⁺ to competitively generate hydroxyl radicals (\cdot OH). Therefore, the Cu²⁺ doped in CCPCA is only used to neutralize excessive GSH (*Science* **375**, 1254-1261 (2022)) in cancer cells and tumors (**Fig. 4i**, **Fig. 7g** and **Fig. 8h**) rather than to generate ROS, and the intratumoral H₂O₂ can be mainly utilized by CAT. Detailed experimental procedures and results are as follows:

First of all, we tested the catalytic activity of CCPCA and free CAT under the same mass of CAT according to the supplier's instructions. One unit (U) of enzyme is defined as the decomposition of 1.0 μ M H₂O₂ per minute at pH 7.0 at 25 °C. The specific activity (SA) was defined as activity units per milligram of enzyme (*Acc. Chem. Res.* **52**, 2190-2200 (2019)).

Materials and method: The CAT (40,000-60,000 units/mg protein, 250 KDa) was purchased from Sigma (CAS: 9001-05-2, SKU: C100-500MG). Catalase Assay Kit purchased from Beyotime Biotechnology was used to detect the CAT enzyme activity and concentration change of H₂O₂.

Supplementary Fig. R1 The catalytic activity of CCPCA and free CAT. (a) Absorption values and corresponding optical photographs after CCPCA treatment at different concentrations. (b) Absorption values and corresponding optical photographs after treatment with different concentration of free CAT. (c) The standard curve of UV-vis absorbance of H₂O₂.

As is shown in **Supplementary Fig. R1**, the SA of CAT (0.5 μ g mL⁻¹) in CCPCA and free CAT were calculated as 45825.3 U/mg protein and 51478.3 U/mg protein according to the standard curve of H₂O₂. The catalytic activity of CAT in CCPCA was 89% of free CAT.

Next, we examined the efficiency of CCPCA and GSH, Cu²⁺, Cu⁺ in the competitive

catalysis of H₂O₂.

Materials and methods: The Cu²⁺ were derived from the extract of CCPCA degraded in pH=1 for 48 h, then the solution was centrifuged with ultrafiltration tube (3.5 kDA) to remove CAT, the filtered solution was adjusted to pH = 6 and the Cu²⁺ was quantified by ICP. Cuprous chloride (CuCl) was weighed in a glove box without oxygen and dissolved in saturated sodium chloride solution to obtain CuCl solution.

a pH=6.0

		1	2	3	4	5	6	7	8	9	10
5 μM	CR	√	√	√	√	√	√	√	√	√	√
50 mM	H ₂ O ₂		√	√	√	√	√	√	√	√	√
1 mM	GSH			√		√				√	
0.05 μg mL ⁻¹	Cu ²⁺				√	√			√	√	
0.05 μg mL ⁻¹	Cu ⁺						√				√
0.60 μg mL ⁻¹	CCPCA							√	√	√	√

Supplementary Fig. R2 CAT within CCPCA catalyzes H₂O₂ much more quickly than Cu²⁺ and Cu⁺. (a) The formulas of the reagents added to each group (1-10). (b) The absorbance value of different test solutions corresponding to (1-10) after 5 min of reaction. (c) The absorbance value of different test solutions corresponding to (1-10) after 30 min of reaction, (n=3). CR: chromogenic reagent in the Catalase Assay Kit. Insert: the optical photograph of different test solutions.

As is shown in **Supplementary Fig. R2**, all test groups with CCPCA were able to catalyze H₂O₂ more rapidly within 5 min, only the Cu⁺ solution could catalyze small amount of H₂O₂ within 30 min.

To verify that CAT in CCPCA competitively catalyzes H₂O₂ substrate and leads to the

inability of Cu^{2+} and Cu^+ to generate $\cdot\text{OH}$, we designed methylene blue (MB) degradation experiment and 3,3',5,5'-tetramethylbenzidine (TMB) oxidation experiment for verification.

Supplementary Fig. R3 CAT within CCPCA leads to the inability of Cu^{2+} and Cu^+ to competitively generate $\cdot\text{OH}$. (a) The formulas of the reagents added to each test tube, MB: methylene blue. (b) The optical photograph of different test solutions after 5 or 30 min of reaction. (c) The absorbance value of different test solutions corresponding to the group of 1-10 in the Supplementary Fig. R3a after 30 min reflection, (n=3). (d) The formulas of the reagents added to each test tube, TMB: 3,3',5,5'-tetramethylbenzidine. (e) The optical photograph of different test solutions after 5 or 30 min of reaction. (f) The absorbance value of different test solutions of 1-10 in the Supplementary Fig. R3d after 30 min reflection, (n=3).

As is shown in **Supplementary Fig. R3**, the addition of Cu^+ (GSH/ Cu^{2+}) can catalyze H_2O_2 to produce $\cdot\text{OH}$ within 30 min, while the test groups treated with CCPCA were able to catalyze H_2O_2 to produce O_2 within 5 min (bubbles in the test tube) with negligible MB degradation or TMB oxidation within 30 min.

In order to further verify our conclusion, we tested the CAT-like activity of CCPCA and POD-like activity of Cu^{2+} and Cu^+ by kinetic analysis respectively:

Supplementary Fig. R4 CAT within CCPCA catalyzes H_2O_2 much more efficiently than Cu^{2+} and Cu^+ . **(a-b)** Kinetic analysis of CAT in CCPCA with CAT-like activity. **(c-d)** Kinetic analysis of Cu^+ with POD-like activity. **(e-f)** Kinetic analysis of Cu^{2+} with POD-like activity.

	CAT	Cu^{2+}	Cu^+
V_{\max} (mM s^{-1})	0.151	3.594	0.716
$[\text{E}]$ (μM)	4×10^{-4}	156.25	156.25
K_m (mM s^{-1})	27.93	74.49	22.6
k_{cat} (s^{-1})	3.77×10^4	23.02	4.57
k_{cat}/K_m ($\text{M}^{-1} \text{s}^{-1}$)	1.34×10^7	3.08×10^2	2.02×10^2

Supplementary Table1. The catalytic efficiency (k_{cat}/K_m) of CAT, Cu^{2+} and Cu^+ .

As is shown in **Supplementary Fig. R4a-f and Table1**, the catalytic efficiency (k_{cat}/K_m) of CAT in CCPCA is much higher than Cu^{2+} and Cu^+ , indicating CAT catalyzes H_2O_2 much more efficiently than Cu^{2+} and Cu^+ , and the intratumoral H_2O_2 can be mainly utilized by CAT.

3. The results of reactive oxygen species in the non-PDT treatment group should be proved because Cu also induces the production of highly reactive oxygen species.

Response: Thank you very much for your comment. According to *Trends Biochem. Sci.* **44**, 401-414 (2019) and *Chem* **5**, 2171-2182 (2019), the redox environment in cancer cells is equilibrium unless the cells are under intense oxidative stress mediated apoptosis signaling pathway. Brisk metabolic or redox adaptation occurs when cancer cells are subjected to metabolism- or ROS-targeted therapy. We have repeated the experiment and obtained the same result. Despite partial of GSH was depleted (Fig. 4i), tumor cells are still able to maintain redox balance without causing intense oxidative stress, as is shown in Fig. 4c, 97.6% and 93.2% of cell viability of A375 cells were recorded in CCPA and CCPCA groups.

4. Immunofluorescence in Figure 7 shows that Ki67 is not located in the nucleus, which may be caused by the non-specific staining.

Response: Thank you very much for your comment. We have replaced these Immunofluorescence Image by immunohistochemistry staining of Ki67 in 4T1 tumors in **Supplementary Fig. 11g**.

Supplementary Fig. 11 g Ki67-stained sections of 4T1 tumors after various treatments.

5. As a pH-responsive Cu-doped calcium phosphate nanoparticles, CCPC degradation in the acidic microenvironment may increase calcium ions in the tumor site. Still, the results show that it does not promote the increase of cytoplasmic calcium ions.

Response: Thank you very much for your comment. The cytoplasmic Ca²⁺ level is low in resting cells, which is maintained at ~100 nM (Cell **131**, 1047-1058 (2007)). In the absence of oxidative stress, calcium ions released by CCPC and CCPA are likely to be transported or stored in the calcium pool of endoplasmic reticulum (*Redox Biol.* **6**, 260-271 (2015), *Chem* **5**, 2171-2182 (2019)). The higher Fluo-4 FL signal in CCPCA may be due to the organelle-selective protein aggregation mediated disturbance of calcium ion regulation caused by the unique ability of high concentration of fluorescent PpIX, which was posited to be a major mechanism by which fluorescent PpIX exerts their cytotoxic effect (*J. Biol. Chem.* **297**, 100778 (2021)). As is shown in the cell experimental results of CCK-8 and flow cytometric analysis in Fig 4a-e, cells treated with CCPCA had the lowest viability than CCPC and CCPA groups after treated with same concentration of nanoparticles.

6. Previous microneedle studies associated with enzymatic reactions/cell incorporation (especially by Zhen Gu group) are suggested to cite, e.g., PNAS, 112 (27) 8260-8265, 2015.

Response: Thank you very much for your valuable comment. According to the reviewer's suggestion, we have cited literatures of Zhen Gu group in the revised manuscript as Ref. 54, Ref. 55, and Ref. 57.

Reviewer #3 (Remarks to the Author): with expertise in NPs and PDT

Comments:

This is an interesting paper in which the authors used a microneedle patch to realize the local treatment of tumor. Specifically, the authors chose 5-ALA and CAT for the in situ synthesis of PpIX and production of oxygen. The experiments have been carefully conducted and the results are encouraging. I would like to recommend the publication of the work in NC after minor revision.

Response: Thank you very much for your positive comment.

Some specific comments:

1. In the introduction section, some related references regarding the use of 5-ALA for PDT and the GSH depletion-enhanced tumor treatment should be mentioned: Adv. Therap. 2019, 2(5), 1800140. ACS Nano 2021, 15(5), 8039–8068.

Response: Thank you very much for your valuable comment. According to the reviewer's suggestion, we have cited these literatures in the revised manuscript as Ref. 30 and Ref. 38.

2. The authors used HA for fabricating the microneedle patch. Please discuss the advantages of using this polymer for the drug loading and release. Could HA be fully degraded in vivo?

Response: Thank you very much for your valuable comment. Hyaluronic acid (HA) is a glycosaminoglycan and is an indigenous component of the connective tissues and dermis. About 50% of the total HA is distributed in the skin. Approximately, one-third proportion of the total HA present in the human body gets degraded on daily basis by **hyaluronidase** and the same amount gets replenished by the freshly synthesized HA (*Carbohydr. Polym.* **267**, 118168 (2021)). According to *Nano Lett.* **16**, 2334-2340 (2016), HA can be used as a local carrier for transdermal administration of drugs owing to its good biocompatibility, biodegradability, strong capacity to pierce and hydrate cuticles, and HA can be fully degraded in vivo to release nanoparticles.

3. The blood biosafety of the patch should be evaluated.

Response: Thank you very much for your valuable comment. The serum biochemical analysis of U87MG tumor-bearing mice in CCPCA *i.t.* and MN-CCPCA treatment groups before and after 30 min laser irradiation were described in the section of “**In vivo personalized theranostics for the U87MG tumor model**” as follows:

“As we know, PpIX can be activated by UV or visible light, causing systemic porphyria and damage to normal tissues⁷. The uncontrolled release of 5-ALA and nonspecific distribution of PpIX by *i.v.* or *i.t.* injections of CCPCA raised biosafety concerns for following irradiation treatment. Although there was no evident damage to major organs of U87MG tumor-bearing mice after *i.v.*, *i.t.* or MN administrations of CCPCA within 14 days (Supplementary Fig. 14), we further conducted serum biochemical analysis to assess the biosafety of *i.t.* or MN administrations of CCPCA before and after laser irradiation. As shown in Fig 8j, there was no significant difference in serum biochemical parameters without laser exposure. However, alanine aminotransferase (ALT) and aspartate aminotransferase (AST) significantly increased in CCPCA *i.t.* group after 30 min laser irradiation. In contrast, there was negligible difference in serum biochemical parameters of liver and kidney function of mice treated with MN-CCPCA patch after 30 min laser irradiation, suggesting that MN administration of CCPCA can improve the biosafety of 5-ALA-PDT”.

Supplementary Fig. 14 H&E staining images of main organs (heart, liver, spleen, lung and kidney) of U87MG tumor-bearing mice after different administrations of 5-ALA and CCPCA for 14 days, scale bar = 100 μ m.

Fig. 8. j Serum biochemical analysis of U87MG tumor-bearing mice in CCPCA *i.t.* and MN-CCPCA groups before and after 30 min laser irradiation, (n = 3). The data represent the means ± SD. ***p < 0.001, **p < 0.01, *p < 0.05, n.s., not significant. Two-tailed Student's t test.

4. Because the authors used Cu, could this element play a role in chemodynamic therapy?

Please discuss.

Response: Thank you very much for your valuable comment. Our experiments showed that the Cu²⁺ doped CCPCA is only used to neutralize excessive GSH (*Science* **375**, 1254-1261 (2022)) in cancer cells and tumors (**Fig. 4i**, **Fig. 7g** and **Fig. 8h**) rather than to generate ROS, and the intratumoral H₂O₂ can be mainly utilized by CAT. The CAT within CCPCA catalyzes H₂O₂ much more efficiently than Cu²⁺ and Cu⁺ in the presence of equal amounts of CAT and Cu (The weight percentages of CAT and Cu in CCPCA are 7.9 wt.% and 6.3 wt.%, respectively. And here we assume that CAT and Cu²⁺ in CCPCA are completely released, resulting in a weight ratio close to 1:1), which leads to the inability of Cu²⁺ and Cu⁺ to competitively generate hydroxyl radicals (-OH). Detailed experimental procedures and results are as follows:

First of all, we tested the catalytic activity of CCPCA and free CAT under the same mass of CAT according to the supplier's instructions. One unit (U) of enzyme is defined as the decomposition of 1.0 μM H₂O₂ per minute at pH 7.0 at 25 °C. The specific activity (SA) was defined as activity units per milligram of enzyme (*Acc. Chem. Res.* **52**, 2190-2200 (2019)).

Materials and method: The CAT (40,000-60,000 units/mg protein, 250 KDa) was purchased from Sigma (CAS: 9001-05-2, SKU: C100-500MG). Catalase Assay Kit purchased from Beyotime Biotechnology was used to detect the CAT enzyme activity and concentration change of H₂O₂.

Supplementary Fig. R1 The catalytic activity of CCPCA and free CAT. (a) Absorption values and corresponding optical photographs after CCPCA treatment at different concentrations. (b) Absorption values and corresponding optical photographs after treatment with different concentration of free CAT. (c) The standard curve of UV-vis absorbance of H_2O_2 .

As is shown in **Supplementary Fig. R1**, the SA of CAT ($0.5 \mu\text{g mL}^{-1}$) in CCPCA and free CAT were calculated as 45825.3 U/mg protein and 51478.3 U/mg protein according to the standard curve of H_2O_2 . The catalytic activity of CAT in CCPCA was 89% of free CAT.

Next, we examined the efficiency of CCPCA and GSH, Cu^{2+} , Cu^+ in the competitive catalysis of H_2O_2 .

Supplementary Fig. R2 CAT within CCPCA catalyzes H_2O_2 much more quickly than Cu^{2+} and Cu^+ . (a) The formulas of the reagents added to each group (1-10). (b) The absorbance

value of different test solutions corresponding to (1-10) after 5 min of reaction. **(c)** The absorbance value of different test solutions corresponding to (1-10) after 30 min of reaction, (n=3). CR: chromogenic reagent in the Catalase Assay Kit. Insert: the optical photograph of different test solutions.

Materials and methods: The Cu^{2+} were derived from the extract of CCPCA degraded in $\text{pH}=1$ for 48 h, then the solution was centrifuged with ultrafiltration tube (3.5 kDA) to remove CAT, the filtered solution was adjusted to $\text{pH} = 6$ and the Cu^{2+} was quantified by ICP. Cuprous chloride (CuCl) was weighed in a glove box without oxygen and dissolved in saturated sodium chloride solution to obtain CuCl solution.

As is shown in **Supplementary Fig. R2**, all test groups with CCPCA were able to catalyze H_2O_2 more rapidly within 5 min, only the Cu^+ solution could catalyze small amount of H_2O_2 within 30 min.

To verify that CAT in CCPCA competitively catalyzes H_2O_2 substrate and leads to the inability of Cu^{2+} and Cu^+ to generate $\cdot\text{OH}$, we designed methylene blue (MB) degradation experiment and 3,3',5,5'-tetramethylbenzidine (TMB) oxidation experiment for verification.

Supplementary Fig. R3 CAT within CCPCA leads to the inability of Cu^{2+} and Cu^+ to competitively generate $\cdot\text{OH}$. **(a)** The formulas of the reagents added to each test tube, MB: methylene blue. **(b)** The optical photograph of different test solutions after 5 or 30 min of reaction. **(c)** The absorbance value of different test solutions corresponding to the group of

1-10 in the Supplementary Fig. R3a after 30 min reflection, (n=3). **(d)** The formulas of the reagents added to each test tube, TMB: 3,3',5,5'-tetramethylbenzidine. **(e)** The optical photograph of different test solutions after 5 or 30 min of reaction. **(f)** The absorbance value of different test solutions of 1-10 in the Supplementary Fig. R3d after 30 min reflection, (n=3).

As is shown in **Supplementary Fig. R3**, the addition of Cu^+ (GSH/ Cu^{2+}) can catalyze H_2O_2 to produce $\cdot\text{OH}$ within 30 min, while the test groups treated with CCPCA were able to catalyze H_2O_2 to produce O_2 within 5 min (bubbles in the test tube) with negligible MB degradation or TMB oxidation within 30 min.

In order to further verify our conclusion, we tested the CAT-like activity of CCPCA and POD-like activity of Cu^{2+} and Cu^+ by kinetic analysis respectively:

Supplementary Fig. R4 CAT within CCPCA catalyzes H_2O_2 much more efficiently than Cu^{2+} and Cu^+ . **(a-b)** Kinetic analysis of CAT in CCPCA with CAT-like activity. **(c-d)** Kinetic

analysis of Cu⁺ with POD-like activity. **(e-f)** Kinetic analysis of Cu²⁺ with POD-like activity.

	CAT	Cu ²⁺	Cu ⁺
V_{max} (mM s ⁻¹)	0.151	3.594	0.716
[E] (μM)	4×10 ⁻⁴	156.25	156.25
K_m (mM s ⁻¹)	27.93	74.49	22.6
k_{cat} (s ⁻¹)	3.77×10 ⁴	23.02	4.57
k_{cat}/K_m (M ⁻¹ s ⁻¹)	1.34×10 ⁷	3.08×10 ²	2.02×10 ²

Supplementary Table1. The catalytic efficiency (k_{cat}/K_m) of CAT, Cu²⁺ and Cu⁺.

As is shown in **Supplementary Fig. R4a-f and Table1**, the catalytic efficiency (k_{cat}/K_m) of CAT in CCPCA is much higher than Cu²⁺ and Cu⁺, indicating CAT catalyzes H₂O₂ much more efficiently than Cu²⁺ and Cu⁺, and the intratumoral H₂O₂ can be mainly utilized by CAT.

5. Please discuss the advantages of this patch by comparing with a hydrogel delivery system that is also loaded with CAT and 5-ALA (and others) as in this work

Response: Thank you very much for your valuable comment. As skin is composed of three layers, epidermis (thickness of 50–150 μm), dermis (thickness of 1–2 mm) and subcutaneous tissue making up a total skin thickness of ~ 3 mm. There are five sublayers of epidermis, including stratum corneum (SC), stratum lucidum, stratum granulosum, stratum spinosum, and stratum basale. SC is the chief barrier of skin composed of stratified, lipid depleted, and protein enriched corneocytes with the thickness of 10–20 μm. For topical hydrogels delivery system, its therapeutic effectiveness is severely limited by the presence of a physiological barrier of SC, which prevents most drugs from entering the skin at therapeutically useful rates. For injectable hydrogels delivery system, there are several disadvantages on biosafety: (1) Pain and needle phobia, leading to poor patient compliance (*Br. J. Anaesth.* **113**, 4-6 (2014)); (2) The possibility of needle-stick injuries, hematoma formation or bleeding and dangerous biological waste and sharp disposal hazard (*J. Control. Release* **260**, 164-182 (2017)). Successful transdermal delivery is based on achieving a suitable balance between effective delivery and safety to the skin (*Nat. Biotechnol.* **26**, 1261-1268 (2008)). To this aim, we

designed fluorescence (FL) imaging experiments in both 4T1 (**Fig. 7**) and U87MG tumor-bearing mice models (**Fig. 8**) to demonstrate the improved bioavailability and biosafety of MN-CCPCA administration comparing to other delivery modes (intravenous injection (*i.v.*), intratumor injection (*i.t.*)) of 5-ALA or CCPCA.

The highlights of this work are to solve the bottlenecks (the heterogeneity of PpIX metabolism and the limitation of physiological synthesis PpIX concentration) of 5-ALA based photodynamic therapy (5-ALA-PDT) for solid tumors through the design of synthetic biology, optimized delivery strategy and real-time imaging guidance. In our work, the phenomenon and molecular mechanism that molecular oxygen can increase the concentration of intratumoral PpIX are found and a synthetic biology approach is developed to break the physiological limitation of PpIX synthesis in different kinds of solid tumors; Besides, by combining a microneedle transdermal delivery strategy and a sustained-release nano-delivery system for specific/local delivery of 5-ALA and CAT, more synthetic PpIX are trapped in tumor cells instead of synthesizing heme, this combination enhances the bioavailability of 5-ALA and CAT and amplifies the signal of FL/PA duplex imaging simultaneously; Finally, Laser irradiation parameters (time point, dosage, duration, and number of laser irradiation) for different solid tumors (such as breast, glioblastoma, melanoma) can be optimized by FL/PA duplex imaging, which overcomes issues of heterogeneous PpIX metabolism in different tumor types and ultimately improves therapeutic efficacy and safety.

REVIEWERS' COMMENTS

Reviewer #1 (Remarks to the Author):

The authors have addressed most of the concerns, and lots of experiments were supplemented in the revised manuscript. Although the work is not sufficiently innovative conceptually, this study's hypothesis and conclusions have now been supported well by the solid evidences provided by the authors and the manuscript is in a better shape when compared with the original version. Nevertheless, there are some claims that need to be clarified before I recommend publication in NC.

1) The definition of synthetic biology used in this study should be described in detail. Catalase catalysis contributed to oxygen production, and enriched the level of intratumoral PpIX, thus catalyst therapy might be more appropriate.

2) The authors supplemented experiment to confirm that CCPCA can suppress the HIF-1 α expression of A375 cells in hypoxic conditions (1% O₂). However, it is strange to find that western blot analysis of HIF-1 α showed a similar gray of protein band under both normoxic and hypoxic conditions. In addition, the improved production of O₂ generation and PpIX production in hypoxic conditions (1% O₂) should be shown in A375 cells after CCPCA treatment.

Reviewer #2 (Remarks to the Author):

The authors have addressed my comments.

Reviewer #3 (Remarks to the Author):

The authors have made great efforts to improve the work and have carefully addressed all my comments and suggestions.

Point-by-point response to Reviewers' comments for manuscript:

“Synthetic biology-instructed transdermal microneedle patch for traceable photodynamic therapy”

- Author responses are in **blue color**.

We would like to Reviewer for all constructive comments and questions on our manuscript.

Reviewer #1 (Remarks to the Author)

Comments:

The authors have addressed most of the concerns, and lots of experiments were supplemented in the revised manuscript. Although the work is not sufficiently innovative conceptually, this study's hypothesis and conclusions have now been supported well by the solid evidences provided by the authors and the manuscript is in a better shape when compared with the original version. Nevertheless, there are some claims that need to be clarified before I recommend publication in NC.

1) The definition of synthetic biology used in this study should be described in detail. Catalase catalysis contributed to oxygen production, and enriched the level of intratumoral PpIX, thus catalyst therapy might be more appropriate.

Response: We thank the reviewer for the thoughtful review and helpful comments. In materials synthetic biology, engineering principles from synthetic biology and materials science are integrated to redesign living systems as dynamic and responsive materials with emerging and programmable functionalities (*Nat. Rev. Mater.* **6**, 332-350 (2021)). First, conceptually, we believe MN-CCPCA meet the requirements of synthetic biology materials. Two substances (5-ALA and CAT) in MN-CCPCA can be used as key elements in regulating PpIX synthesis. The CAT affects the activity of PpIX-related metabolic enzymes by O₂

generation, and 5-ALA is the necessary precursor for PpIX synthesis. The MN patch and CCPCA delivery platforms were used to achieve the synergistic effect of the two to maximize the physiological synthesis concentration of PpIX. Secondly, the experimental results also verify that the anabolic-based PpIX regulation is effective. After the application of MN-CCPCA, the synthetic concentration of PpIX in different types of tumors is higher than that of the same amount of free 5-ALA administrations (i.v., i.t. and MN), and the tumor retention time of PpIX is longer, which allows the challenges (short treatment time window and PpIX concentration) based on 5-ALA-PDT to be solved. However, catalytic therapy generally refers to that the chemical substances produced by catalysis have a direct therapeutic effect on the organisms (*Adv. Mater.* 31, 1901778 (2019)). This manuscript does not mention the therapeutic effect directly produced by O₂ generation, such as the inhibitory effect differences between MN-CCP and MN-CCPC on A375 tumors (Fig. 6c-e). In fact, the catalytic process of H₂O₂ to generate O₂ by CAT is only the beginning of the entire metabolic regulation. The combined effect of 5-ALA and O₂ on PpIX synthesis in tumors makes it possible to regulate PpIX synthesis through the HIF-1 α -FECH-ALAS axis. Therefore, the concept of synthetic biology is more suitable for our manuscript.

2) The authors supplemented experiment to confirm that CCPCA can suppress the HIF-1 α expression of A375 cells in hypoxic conditions (1% O₂). However, it is strange to find that western blot analysis of HIF-1 α showed a similar gray of protein band under both normoxic and hypoxic conditions. In addition, the improved production of O₂ generation and PpIX production in hypoxic conditions (1% O₂) should be shown in A375 cells after CCPCA treatment.

Response: We thank the reviewer for the thoughtful review and helpful comments. In Fig. 3I, although the gray seems similar at 0 h (pre-treatment), the expression amounts of HIF-1 α at 6, 12 and 24 h (after treat with 80 μ g mL⁻¹ CCPCA NPs) under normoxic condition are all noticeably higher than those under hypoxic condition. The expression amount of HIF-1 α in A375 cells under 21% O₂ was almost completely degraded for 12 or 24-h treatment, while there was still a small amount of HIF-1 α protein expression in A375 cells under 1% O₂ condition at 12, 24 h. The same phenomenon appeared in A375 cells after treatment of

different concentrations of CCPCA NPs for 24 h (Fig. 3m). Therefore, the expression of HIF-1 α in CCPCA-treated A375 cells was different under normoxic and hypoxic conditions.

More importantly, tumors are inherently hypoxic microenvironments, and we believe that the in vivo data are more convincing than the cellular level results. In Fig. 5a-e, compared to the CCPA group, 2.4-fold higher sO₂ level in tumors was observed in the MN-CCPCA NP group at 24 h post-treatment. We also demonstrated that the PpIX fluorescence intensity obtained by MN-CCPCA treatment was 2.2-fold higher than that obtained by MN-CCPA treatment at the METP of PpIX. These results indicating that 5-ALA and CAT co-delivery could promote PpIX accumulation in A375 tumors.

Reviewer #2 (Remarks to the Author)

Comments:

The authors have addressed my comments.

Response: Thank you very much for your positive comments.

Reviewer #3 (Remarks to the Author)

Comments:

The authors have addressed my comments.

The authors have made great efforts to improve the work and have carefully addressed all my comments and suggestions.

Response: Thank you very much for your positive comments.